# Approximation with CNNs in Sobolev Space: with Applications to Classification

**Guohao Shen**[*]
Department of Applied Mathematics, The Hong Kong Polytechnic University
Hung Hom, Kowloon, Hong Kong SAR, China
guohao.shen@polyu.edu.hk

**Yuling Jiao**[*]
School of Mathematics and Statistics
and Hubei Key Laboratory of Computational Science
Wuhan University, Wuhan 430072, China
yulingjiaomath@whu.edu.cn

**Yuanyuan Lin**[†]
Department of Statistics, The Chinese University of Hong Kong
Shatin, New Territories, Hong Kong SAR, China
ylin@sta.cuhk.edu.hk

**Jian Huang**[†]
Department of Applied Mathematics, The Hong Kong Polytechnic University
Hung Hom, Kowloon, Hong Kong SAR, China
j.huang@polyu.edu.hk

## Abstract

We derive a novel approximation error bound with an explicit prefactor for Sobolev-regular functions using deep convolutional neural networks (CNNs). The bound is non-asymptotic in terms of the network depth and filter lengths, in a rather flexible way. For Sobolev-regular functions which can be embedded into the Hölder space, the prefactor of our error bound depends on the ambient dimension polynomially instead of exponentially as in most existing results, which is of independent interest. We also establish a new approximation result when the target function is supported on an approximate lower-dimensional manifold. We apply our results to establish non-asymptotic excess risk bounds for classification using CNNs with convex surrogate losses, including the cross-entropy loss, the hinge loss, the logistic loss, the exponential loss and the least squares loss. We show that the classification methods with CNNs can circumvent the curse of dimensionality if input data is supported on a neighborhood of a low-dimensional manifold.

## 1 Introduction

Classification methods with hypothesis spaces specified through deep neural networks have achieved remarkable successes in a variety of machine learning tasks [36]. In particular, *convolutional neural*

---

[*]Guohao Shen and Yuling Jiao contributed equally to this work
[†]Corresponding authors

36th Conference on Neural Information Processing Systems (NeurIPS 2022).

*networks* (CNNs) have demonstrated outstanding performance in many applications, including computer vision [35], natural language processing [57], and sequence analysis in bioinformatics [1, 65]. However, to the best of our knowledge, there have only been limited studies on the approximation properties with CNNs and non-asymptotic excess risk bounds for various classification methods using CNNs.

## 1.1 Related work

There has been much effort devoted to understanding the approximation properties of deep neural networks in recent years. Many interesting results have been obtained concerning the approximation power of deep neural networks for multivariate functions; some examples of important recent works include [59, 60, 40, 49, 53, 54, 44, 13, 28]. These works focused on the approximation power for ReLU activated feedforward neural networks (FNNs) on different kinds of smooth functions.

The approximation power of CNNs has been studied in limited works. [2] studied the approximation power of the composited CNNs, where the network consists of fully-connected CNNs followed by a fully-connected layer. They proved that under suitable conditions, convolution neural networks can inherit the universal approximation property of its last fully connected layers. [48] showed that for translation equivariant functions, all upper and lower approximation bounds are equivalent between FNNs and CNNs with the same order of network length and size. [47] showed that ResNet-type CNNs, a special CNN architecture with skip-layer connections (or identity connections between inconsecutive layers), can replicate the learning ability of FNNs having block-sparse structures. The ResNet-type CNN in [47], as claimed, can be dense, and its width, channel size, and filter size are constant with respect to sample size rather than the diverging one in [48].

In addition, a series of papers studied the approximation power of deep CNNs with Toeplitz type convolutional matrices [66, 64, 63]. The universality of such CNNs was established in [64] for target functions restricted to the Sobolev space $W^{\beta,2}(\mathbb{R}^d)$ with $\beta \geq d/2 + 2$. The approximation error is of the order $O(L^{-1/2-1/d})$ (up to a logarithmic factor), where $L$ is the number of layers of such CNNs. But the CNN defined in [64] has width increasing linearly with respect to depth. To overcome the difficulty in theoretical analysis of such CNNs, [63] introduced a downsampling operator to reduce the widths and the approximation power of downsampled CNNs for a special class of functions ($\beta$-Hölder continuous ridge function) was studied. The approximate rate is shown to be $O(N^{-\beta})$ by downsampled CNNs with a uniform filter length $s = O(4N + 6)$ and width $\mathcal{W} = O(N)$ for positive integer $N$. Further, it was shown that a downsampled CNN can compute the same function as a FNN does, with the total number of free parameters of downsampled CNN being at most 8 times of that of FNN. For the CNNs considered by [66, 16, 63] and [64], each convolutional layer has only one filter with size $s$ and the weight matrix is of a special Toeplitz type with shape $(d_{in} + s) \times d_{in}$ and no fully-connected layer is allowed in hidden layers. Approximation properties of convolutional architectures for target functions defined on infinite time domains tailored to temporal sequence data were studied in [27].

To mitigate the curse of dimensionality and show the advantages of CNNs in classification problems, several works studied the approximation as well as the statistical learning theories under some low-dimensional assumptions on the target function or the data distribution. In [32, 33], the posterior probability function is assumed to be Hölder smooth satisfying hierarchical max-pooling model. In [34], the target function is assumed to be spatially rotation invariant. In [18] and [38], input data is assumed to be supported on spheres and manifolds, respectively. Under different structural assumptions on the target function, similar results have also been proved for FNNs [7, 51, 52]. If data is assumed to be supported on low-dimensional manifolds, the curse of dimensionality can also be mitigated for FNNs [50, 44, 12, 28].

## 1.2 Our contributions

We study the approximation power of CNNs for functions in Sobolev spaces, including the special case of Hölder class and apply our new approximation results to establish non-asymptotic error bounds for several important classification methods using CNNs. Our main contributions are as follows.

(i) We derive a novel error bound for approximating Sobolev-regular functions using deep CNNs, with the error bound explicitly expressed in terms the network parameters and model parameters.

The error bound depends on the network depth and filter lengths in a rather flexible way. For functions which can be embedded into Hölder space, the prefactor of our error bound depends on the ambient dimension polynomially instead of exponentially as in existing results. We also establish a novel approximation result when the target function is supported on an approximate lower-dimensional manifold, which shows that CNNs are capable of mitigating the curse of dimensionality under such a distributional condition.

(ii) As an application of our CNN approximation results, we establish non-asymptotic bounds for the stochastic and approximation errors of classification methods with a general class of convex surrogate losses in Sobolev space using CNNs. Our main results are also applicable to other problems that use CNNs approximations.

(iii) We apply our general results to establish the non-asymptotic excess risk bounds for classification using CNNs with convex surrogate losses in Sobolev space, including the cross-entropy loss, the hinge loss (SVM), the logistic loss, the exponential loss, and the least squares loss. We show that classification methods with CNNs can circumvent the curse of dimensionality if the input data is supported on a neighborhood of a low-dimensional manifold embedded in the ambient space.

Table 1: A comparison of some recent convolutional neural network approximation results.

| | Network | Target function | Flexible filter length | Explicit prefactor | Low-dimensional Result |
|---|---|---|---|---|---|
| [48] | CNN | FNN | ✗ | ✗ | ✗ |
| [47] | ConvResNet | FNN | ✓ | ✗ | ✗ |
| [64] | CNN | Sobolev | ✓ | ✗ | ✗ |
| [33] | CNN | Hölder | ✗ | ✗ | ✓ |
| [38] | ConvResNet | Besov | ✓ | ✗ | ✓ |
| This paper | CNN | Sobolev and Hölder | ✓ | ✓ | ✓ |

In Table 1, we summarize and compare some recent and most related work on CNN approximation results for various function spaces.

The statistical convergence properties of the excess risk of the empirical risk minimizer (ERM) for the misclassification 0-1 loss have been studied extensively under various conditions. Different types of hypothesis spaces have been used in these approaches, including the linear space, the reproducing kernel Hilbert space and the class of tree-based models, see, for example, [9, 55, 42, 6, 29, 8] and [62]. However, there are limited studies worked on deep binary classifications by FNNs [46, 25, 30], ResNet [26, 45] and CNNs [32, 33, 38].

## 2 Approximation power of CNNs

### 2.1 Convolutional neural networks

Convolutional networks are a special type of structured sparse feedforward neural network (FNN) that use convolution in place of general matrix multiplication in at least one of their layers [21]. There are different formulations of CNNs in the literature [2, 66, 47, 37, 63, 64, 33, 32, 38]. In this paper, we consider ReLU activated downsampled CNNs with bias vectors in the convolutional layers defined in [63]. We do not require the norm of network parameters (weight and bias) to be uniformly bounded.

We consider a general CNN function $f_{\mathrm{CNN}} : \mathcal{X} \to \mathbb{R}$, where $\mathcal{X} \subset \mathbb{R}^d$. For simplicity, we take $\mathcal{X} = (0,1)^d$. Suppose $f_{\mathrm{CNN}}$ has $L$ number of hidden layers, then it can be expressed as

$$f_{\mathrm{CNN}}(x) = A_{L+1} \circ A_L \circ \cdots \circ A_2 \circ A_1(x), x \in \mathcal{X}, \tag{1}$$

where $\circ$ denotes the functional composition. The $A_i$'s are either convolutional operators or down-sampling operators. For convolutional layers, $A_i(x) = \sigma(W_i^c x + b_i^c)$, where $W_i^c \in \mathbb{R}^{d_i \times d_{i-1}}$ is the structured sparse Toeplitz type weight matrix induced by the convolutional filter $\{w_j^{(i)}\}_{j=0}^{s^{(i)}}$ with filter length $s^{(i)} \in \mathbb{N}^+$, and $b_i^c \in \mathbb{R}^{d_i}$ is a bias vector, and $\sigma$ is the rectified linear unit (ReLU) activation function applying to each component of the input vector.

For downsampling layers, for any $x \in \mathbb{R}^{d_{i-1}}$, $A_i(x) = D_i(x) = (x_{jm_i})_{j=1}^{\lfloor d_{i-1}/m_i \rfloor}$, where $D_i : \mathbb{R}^{d_i \times d_{i-1}}$ is the downsampling operator with scaling parameter $m_i \leq d_{i-1}$ in the $i$-th layer. We introduce the convolutional operation and downsampling operation [63] in details in Appendix **??**.

We focus on the function class consisting of deep CNNs, denoted by $\mathcal{F}_{\text{CNN}}$ which is defined as

$$\mathcal{F}_{\text{CNN}} = \{ f_{\text{CNN}} \text{ defined in (1) over all possible choice of } A_i, i = 1, \ldots, L+1 \}. \tag{2}$$

For a general function class $\mathcal{F}$ and any $f \in \mathcal{F}$, define the $\| \cdot \|_\infty$ metric on $\mathcal{F}$ by $\sup_{x \in \mathcal{X}} \|f(x)\|_\infty$. Then let $\mathcal{B} := \sup_{f \in \mathcal{F}_{\text{CNN}}} \|f\|_\infty$ denote the bound of the functions in $\mathcal{F}_{\text{CNN}}$. Let $L$ be the number of hidden layers and $\mathcal{S}$ be the total number of parameters for networks in $\mathcal{F}_{\text{CNN}}$ and let $s_{\min}$ and $s_{\max}$ be the minimum and maximum filter length over convolutional layers respectively.

## 2.2 Approximation in Sobolev space

We use the Sobolev and Hölder classes for the target functions in our approximation results since they are well-established formulations for describing smooth functions in the literature. Indeed, earlier works on approximation theory of neural networks have been developed for Sobolev functions [59, 64, 24] and for Hölder functions [33, 51].

Let $\mathbb{N}_0$ be the set of non-negative integers and $\beta \in \mathbb{N}_0$. The Sobolev class of functions $W^{\beta,p}(\mathcal{X})$ with $\mathcal{X} = (0,1)^d$ is defined as

$$W^{\beta,p}(\mathcal{X}) = \left\{ f \in L^p(\mathcal{X}) : D^\alpha f \in L^p(\mathcal{X}) \text{ for all } \alpha \in \mathbb{N}_0^d \text{ with } \|\alpha\|_1 \leq \beta \right\}, \tag{3}$$

where $1 \leq p \leq \infty$, $D^\alpha = \partial^{\alpha_1} \cdots \partial^{\alpha_d}$ with $\alpha = (\alpha_1, \ldots, \alpha_d)^\top \in \mathbb{N}_0^d$, and $\|\alpha\|_1 = \sum_{i=1}^d \alpha_i$. For $f \in W^{\beta,p}(\mathcal{X})$ and $0 \leq m \leq \beta$, we define the norm $\|f\|_{W^{m,p}(\mathcal{X})} := \left( \sum_{0 \leq \|\alpha\|_1 \leq m} \|D^\alpha f\|_{L^p(\mathcal{X})}^p \right)^{1/p}$ for $1 \leq p < \infty$, and define $\|f\|_{W^{m,\infty}(\mathcal{X})} := \max_{0 \leq \|\alpha\|_1 \leq m} \|D^\alpha f\|_{L^\infty(\mathcal{X})}$.

In the remainder of the paper, for any positive integers $M, N \in \mathbb{N}^+$, let $\mathcal{F}_{\text{CNN}}$ be the class of CNNs defined in (2) with depth $L$, size $\mathcal{S}$ and filter length specified as follows:

$$L \leq 42(\lfloor \beta \rfloor + 1)^2 M \lceil \log_2(8M) \rceil \lceil \frac{\mathcal{W} - 1}{s_{\min} - 1} \rceil, \tag{4}$$

$$2 \leq s_{\min} \leq s_{\max} \leq \mathcal{W}, \tag{5}$$

$$\mathcal{S} \leq 8 \mathcal{W} L, \tag{6}$$

where

$$\mathcal{W} = 38^2 (\lfloor \beta \rfloor + 1)^4 d^{2\lfloor \beta \rfloor + 2} N^2 \lceil \log_2(8N) \rceil^2, \tag{7}$$

denotes the maximum incremental width (number of neurons) for consecutive layers in the network and $\lceil a \rceil$ denotes the smallest integer no less than $a$.

**Theorem 2.1.** *Assume that $f \in W^{\beta,p}(\mathcal{X})$ with $1 \leq \beta \in \mathbb{N}_0$, $1 \leq p \leq \infty$ and $\|f\|_{W^{\beta,p}(\mathcal{X})} \leq B_0$ for $B_0 > 0$. For any $M, N \in \mathbb{N}^+$, there exists a function $f_{CNN} \in \mathcal{F}_{CNN}$ defined in (2) with depth $L$ and filter lengths and size $\mathcal{S}$ specified in (4), (5) and (6), such that for $m = 0, 1$,*

$$\|f - f_{CNN}\|_{W^{m,p}(\mathcal{X})} \leq C_0(d, \beta, p)(NM)^{-2(\beta - m)/d},$$

*where $C_0(d, \beta, p) = 37 \cdot 2^{2\beta + 2d/p} B_0^2 (\beta + 1)^3 \times \{\pi^{-d/2} \Gamma(d/2 + 1)\}^{2/p+1} (1 + 2\sqrt{d})^d d^{4\beta}$. Here $\Gamma(\cdot)$ is the gamma function.*

Our result is new for approximating Sobolev-regular functions in $W^{\beta,p}(\mathcal{X})$ for a positive integer $\beta \in \mathbb{N}^+$ with respect to the Sobolev norms $\| \cdot \|_{W^{m,p}(\mathcal{X})}$ for $m = 0, 1$ in terms of explicitly defined approximation error prefactor, clearly defined and flexible network parameters, as well as an fast approximation rate comparable with [22]. A toy example is provided in Appendix D to illustrate the approximation power of CNNs and examine how the approximation error varies according to the filter size and depth of the networks.

In [24], an approximation result on $\beta$-smooth functions by networks with squared ReLU activation function is also provided under the norm $\| \cdot \|_{W^{m,p}(\mathcal{X})}$ for positive integer $m < \beta$. In [22], the

approximation rates of ReLU DNN for Sobolev-regular functions with respect to the weaker Sobolev norms $\| \cdot \|_{W^{s,p}(\mathcal{X})}$ for $s \in [0, 1]$ were analyzed, which can be seen as a generalization of the result in [59]. Approximation results on Sobolev-regular functions $W^{\beta,p}(\mathcal{X})$ by networks with general activation functions are derived in [23], where the error is with respect to $\| \cdot \|_{W^{m,p}(\mathcal{X})}$ for some integer $m < \beta$.

By the general Sobolev inequality, see Theorem 6 in Chapter 5 of [15], $W^{k,p}(\mathcal{X})$ with $kp > d$ can be embedded into a Hölder class $\mathcal{H}^\beta(\mathcal{X}, B_0)$ (with $\beta = k - \lfloor d/p \rfloor - 1$) defined as

$$\mathcal{H}^\beta(\mathcal{X}, B_0) = \left\{ f : \mathcal{X} \to \mathbb{R}, \max_{\|\alpha\|_1 \leq \lfloor \beta \rfloor} \|D^\alpha f\|_\infty \leq B_0 \text{ and } \max_{\|\alpha\|_1 = s} \sup_{x \neq y} \frac{|D^\alpha f - D^\alpha f(x)|}{\|x - y\|_2^r} \leq B_0 \right\}. \tag{8}$$

where $\beta > 0$ and $\lfloor \beta \rfloor \in \mathbb{N}_0$ denotes the largest integer strictly smaller than $\beta$. Because of the extra regularity on the Hölder class, a much improved prefactor $C_0$ in the error bound of Theorem 2.1 can be obtained.

**Theorem 2.2.** *Let $f \in \mathcal{H}^\beta(\mathcal{X}, B_0)$ be defined in (8) and let $X \in \mathbb{R}^d$ be a random vector whose probability distribution is supported on $\mathcal{X} = (0, 1)^d$ and absolutely continuous with respect to the Lebesgue measure. For any $M, N \in \mathbb{N}^+$, there exists a function $f_{CNN} \in \mathcal{F}_{CNN}$ defined in (2) with depth $L$ and filter lengths and size $\mathcal{S}$ specified in (4), (5) and (6), such that*

$$\mathbb{E}|f(X) - f_{CNN}(X)| \leq C_0(d, \beta)(NM)^{-2\beta/d},$$

*where $C_0(d, \beta) = 18B_0(\beta + 1)^2 d^{\beta + (\beta \vee 1)/2}$. Here $a \vee b := \max\{a, b\}$ for $a, b \in \mathbb{R}$.*

The convergence rate $(NM)^{-2\beta/d}$ in Theorems 2.1 and 2.2 with respect to the network depth and filter lengths specified by $M$ and $N$, is in line with the nearly optimal rate of ReLU FNNs on smooth functions under Sobolev norm in [24] and on Hölder smooth functions under $L^\infty$ norm in [28]. The prefactor $18B_0(\beta + 1)^2 d^{\beta + (\beta \vee 1)/2}$ of approximation error depends on the dimension $d$ polynomially, different from the exponential dependence in many existing neural network approximation results mentioned in Section 1.1.

## 2.3 Approximation with a lower-dimensional support

In many modern machine learning problems, the ambient dimension $d$ of the input data could be very large, which results in an extremely slow convergence rate. This fact is known as the *curse of dimensionality*. Fortunately, many types of data have a low-dimensional latent structure, that is, although the ambient dimension $d$ is large, the distribution of the data is approximately supported on a low-dimensional subset of $\mathbb{R}^d$, in which case the approximation error bound can be substantially improved. We establish an approximation result for CNNs in Sobolev spaces under a low-dimensional support assumption.

**Assumption 2.3.** The distribution of $X$ is supported on $\mathcal{M}_\rho$, a $\rho$-neighborhood of $\mathcal{M} \subset \mathcal{X}$, where $\mathcal{M}$ is a compact $d_{\mathcal{M}}$-dimensional Riemannian submanifold and $\mathcal{M}_\rho = \{x \in \mathcal{X} : \inf\{\|x - y\|_2 : y \in \mathcal{M}\} \leq \rho\}$ for $\rho \in (0, 1)$.

In real-world applications, data are hardly observed to locate on an *exact manifold*, instead they could be more realistically viewed as consisting of a latent part supported on a low-dimensional manifold $\mathcal{M}$ plus noises. Therefore, Assumption 2.3 is more realistic than the exact manifold assumption [50, 12, 38].

Define

$$d_\varepsilon = O(d_{\mathcal{M}} \varepsilon^{-2} \log(d/\varepsilon)), \; \varepsilon \in (0, 1), \tag{9}$$

$$\rho_\varepsilon = C_2(NM)^{-2\beta/d_\varepsilon}(\beta + 1)^2 d^{1/2} d_\varepsilon^{3\beta/2} \times [(d/d_\varepsilon)^{1/2} + 1 - \varepsilon]^{-1}(1 - \varepsilon)^{1-\beta}. \tag{10}$$

In our error bound results below, it is the $d_\varepsilon$ that will affect the convergence rate. Since it is often the case that $d_{\mathcal{M}} \ll d$ and therefore $d_\varepsilon \ll d$, the manifold assumption will lead to a better convergence rate than those in Theorems 2.1 and 2.2. In addition, we require $\rho \leq \rho_\varepsilon$.

**Theorem 2.4.** *Suppose that Assumption 2.3 holds, $f \in \mathcal{H}^\beta(\mathcal{X}, B_0)$ and the distribution of $X$ is absolutely continuous with respect to the Lebesgue measure. Let $d_\varepsilon$ and $\rho_\varepsilon$ be defined in (9)*

and (10), respectively. Assume that $\rho \leq \rho_\varepsilon$. Then, for any $M, N \in \mathbb{N}^+$, there exists a CNN function $f_{CNN} \in \mathcal{F}_{CNN}$ with depth $L$ and filter lengths and size $\mathcal{S}$ specified in (4), (5) and (6) with $\mathcal{W} = 38^2(\lfloor \beta \rfloor + 1)^4 d_\varepsilon^{2\lfloor \beta \rfloor + 2} N^2 \lceil \log_2(8N) \rceil^2$ such that

$$\mathbb{E}|f(X) - f_{CNN}(X)| \leq C(d, \beta)(NM)^{-2\beta/d_\varepsilon},$$

where $C(d, \beta) = (18 + C_2)B_0(1 - \varepsilon)^{-\beta}(\beta + 1)^2 d^{1/2} d_\varepsilon^{3\beta/2}$.

Since the intrinsic dimension $d_\varepsilon$ can be much smaller than the ambient dimension $d$, the rate $(NM)^{-2\beta/d_\varepsilon}$ in Theorem 2.4 is greatly improved compared with the rate $(NM)^{-2\beta/d}$ in Theorem 2.2 and the curse of dimensionality is mitigated. The result here is of independent interest and can be useful in other problems that involve the use of CNNs.

## 3   Excess risk in classification

In this section, we present the application of the approximation results Theorems 2.1 and 2.4 to the error analysis of classification with CNNs.

Consider a binary classification problem with a predictor $X \in \mathcal{X} \subset \mathbb{R}^d$ and its binary label $Y \in \{1, -1\}$. We are interested in learning a classifier $h : \mathcal{X} \to \{1, -1\}$ from a class of functions, or a hypothesis space, denoted by $\mathcal{H}$. Let the joint distribution of $(X, Y)$ be $\mathbb{P}$. The goal is to find a classifier that minimizes the misclassification error or the 0-1 risk: $R_*(h) = \mathbb{P}\{h(X) \neq Y\}, h \in \mathcal{H}$. Denote the misclassification risk minimizer at the population level by $h_0 = \operatorname{argmin}_{h \text{ measurable}} R_*(h)$. For any $h \in \mathcal{H}$, the excess risk of $h$ is $R_*(h) - R_*(h_0)$, the difference between the misclassification errors of $h$ and $h_0$.

Since the probability measure $\mathbb{P}$ is unknown in practice, the classifier $h$ will be learned based on a random sample $S = \{(X_i, Y_i)\}_{i=1}^n$ from $\mathbb{P}$, where $n$ is the sample size. The empirical risk minimizer (ERM) is defined by

$$\hat{h}_n \in \operatorname*{argmin}_{h \in \mathcal{H}} \frac{1}{n} \sum_{i=1}^n \mathbb{1}(h(X_i) \neq Y_i). \tag{11}$$

However, the empirical risk function based on the 0-1 loss is non-continuous and non-convex, thus this minimization problem is typically computationally intractable.

Rather than minimizing the non-smooth 0-1 loss, many popular methods adopt a proper convex loss function to train classifiers with computational efficiency that can be done in polynomial time. Moreover, proper surrogate convex loss functions have been shown to be consistent with the 0-1 loss function for binary classification problems [61, 5].

### 3.1   Convex surrogate loss functions

Let $\phi$ be a given convex univariate loss function $\phi : \mathbb{R} \to [0, \infty)$. We consider the risk function with respect to the loss $\phi$

$$R(f) \equiv R^\phi(f) = \mathbb{E}\{\phi(Yf(X))\}, \tag{12}$$

where for simplicity of notation and without causing confusion we omit the superscript $\phi$ in $R$, similarly for $\hat{f}_n$ and $f_0$ defined below. For a given random sample $S = \{(X_i, Y_i)\}_{i=1}^n$, we denote the ERM over $\mathcal{F}_{CNN}$ with a given loss $\phi$ by

$$\hat{f}_n \in \arg\min_{f \in \mathcal{F}_{CNN}} \frac{1}{n} \sum_{i=1}^n \phi(Yf(X_i)). \tag{13}$$

Based on the ERM $\hat{f}_n$, a classifier $\hat{h}_n(x) := \operatorname{sign}(\hat{f}_n(x))$ for $x \in \mathcal{X}$ can be defined. As shown in [61, 5], for a properly chosen $\phi$, $\hat{f}_n$ can help reduce the excess risk $R_*(\hat{h}_n) - R_*(h_0)$. Specifically, define the measurable minimizer of $R$ in (12) as

$$f_0 = \operatorname*{argmin}_{f \text{ measurable}} \mathbb{E}\{\phi(Yf(X))\}, \tag{14}$$

and the corresponding minimal $\phi$-risk as $R_0 = R(f_0)$. Then for a proper $\phi$, we have $\psi(R_*(\hat{h}_n) - R_*(h_0)) \leq R(\hat{f}_n) - R(f_0)$, where $\psi : [-1, 1] \to [0, \infty)$ is a nonnegative continuous function, invertible on $[0, 1]$, and achieves its minimum at 0 with $\psi(0) = 0$. A variety of classification methods are based on this tactic. Generally, classification-calibrated $\phi$ considered in this paper is non-increasing and convex; details are given in the Supplementary Materials.

Define the conditional probability

$$\eta(x) = \mathbb{P}(Y = 1 \mid X = x), \ x \in \mathcal{X}. \tag{15}$$

Let the conditional $\phi$-risk of $f$ given $X = x$ be denoted by $R_f(x) := \mathbb{E}\{\phi(Yf(X)) \mid X = x\}$. SVM and cross entropy, two important examples of $\phi$, and the corresponding $f_0$, $R(f_0)$, $\psi$ and its inverse on $[0, 1]$ in Table 2 in the appendix. For the form of $\psi$, Theorem 34 in [11] shows that if $\phi$ is convex, $\phi''(0) > 0$ exists and $\phi' < 0$, then $\psi(u) = u^2$.

Table 2: Minimizer and minimal conditional risk under SVM and cross entropy loss functions $\phi$. Bound $\phi_B$ of $\phi$, Lipschitz constant $B_\phi$ and $\Delta_\phi(T)$ for the truncated $f_0$ under SVM hinge loss function restricted to $[-\mathcal{B}, \mathcal{B}]$ for $1 \leq T \leq \mathcal{B}$ and cross entropy loss function restricted to $[-\mathcal{B}, \mathcal{B}]$ for $T \leq \mathcal{B} < 0.5$.

|  | SVM | Cross entropy |
| --- | --- | --- |
| $\phi(a)$ | $\max\{1 - a, 0\}$ | $-\log\{0.5 + a\}$ |
| $f_0(x)$ | $\text{sign}(2\eta - 1)$ | $\eta - 0.5$ |
| $R_{f_0}(x)$ | $1 - \|2\eta - 1\|$ | $-\eta\log(\eta) - (1 - \eta)\log(1 - \eta)$ |
| $\psi(\theta)$ | $\|\theta\|$ | $\theta^2$ |
| $\phi_B$ | $\mathcal{B} + 1$ | $-\log\{0.5 - \mathcal{B}\}$ |
| $B_\phi$ | ** | $1/(0.5 - \mathcal{B})$ |
| $\Delta_\phi(T)$ | 0 | $-\log\{1 + (T - 0.5)\}$ |

Note: The conditional probability $\eta(x)$ defined in (15) is written as $\eta$ for notational simplicity. "**" stands for "does not exist"

# 4 Non-asymptotic error bounds

For the ERM $\hat{f}_n$ defined in (13), we first state a basic inequality for bounding the excess risk of $\hat{f}_n$.

**Lemma 4.1.** *For any loss $\phi$ and any random sample $S = \{(X_i, Y_i)\}_{i=1}^n$, the excess $\phi$-risk of the ERM $\hat{f}_n$ satisfies $R(\hat{f}_n) - R(f_0) \leq StoErr + AppErr$, where*

$$StoErr = 2 \sup_{f \in \mathcal{F}} |R(f) - R_n(f)|, \ AppErr = \inf_{f \in \mathcal{F}} R(f) - R(f_0). \tag{16}$$

Therefore, the excess risk of $\hat{f}_n$ is bounded by the sum of two terms: the stochastic error and the approximation error. For a given loss function $\phi$, the upper bound no longer depends on the ERM $\hat{f}_n$, but the function class $\mathcal{F}$ and the random sample $S$. The stochastic error in (16) depends on the complexity of $\mathcal{F}$ and the approximation error in (16) depends on the approximation power of the class $\mathcal{F}$ for $f_0$.

## 4.1 Stochastic error

We bound the stochastic error in terms of the pseudo-dimension of the downsampled CNNs defined in (2); particularly, we further bound its pseudo-dimension and express the bound in terms of quantities related to CNNs.

**Theorem 4.2.** *[Stochastic error bound] Suppose that $\phi$ is convex and non-increasing. For any $M, N \in \mathbb{N}^+$, let $\mathcal{F}_{CNN}$ be the class of CNNs defined in (2) with $\mathcal{B} \geq 0$, depth $L$ and size $\mathcal{S}$ and let $\phi_B := \sup_{|a| \leq \mathcal{B}} \phi(a)$. Then, for any $\delta \in (0, 1)$, with probability at least $1 - \delta$, the stochastic error in (16) satisfies*

$$StoErr \leq \frac{2\phi_B}{\sqrt{n}} \left( C_0 \sqrt{\mathcal{S}L \log(\mathcal{S}) \log(n)} + \sqrt{\log(1/\delta)} \right),$$

*where $C_0 > 0$ is a universal constant.*

The quantity $\phi_B$ for common loss functions are presented in Table 2. It worth noting that the error bound here does not require the norm of the CNN parameters (of weight and bias) to be uniformly bounded. In comparison, those stochastic error bounds based on the covering number generally assume an uniformly bounded norm on CNN parameters, which may hinder the approximation power of network since most approximation results need the norm of network parameters to tend to infinity as the approximation error tends to zero.

## 4.2 Approximation error

We derive an upper bound of the approximation error by relating $\inf_{f \in \mathcal{F}_{\mathrm{CNN}}} R(f) - R(f_0)$ to $\inf_{f \in \mathcal{F}_{\mathrm{CNN}}} \|f - f_0\|$ under proper conditions on $\phi$ and $\eta$, where $R$ is defined in (12). The target function $f_0$ may be non-smooth or unbounded, which poses extra difficulty. Most of the existing studies on the approximation properties of networks assume smooth and bounded target functions. As shown in Table 2, the target function $f_0$ for SVM is non-continuous; for the cross entropy loss, the loss is unbounded on $[-0.5, 0.5]$. For logistic loss and exponential loss, we have $f_0(x) \in \{+\infty, -\infty\}$ once $\eta(x) \in \{0, 1\}$. We overcome these difficulties by imposing mild conditions on the conditional probability $\eta$ and the loss function $\phi$, and use the truncation technique to analyze the approximation.

**Assumption 4.3.** (a) The conditional probability $\eta(x) = \mathbb{P}\{Y = 1 \mid X = x\}$ is continuous on the support of $\mathcal{X}$ and the probability measure of $X$ is absolutely continuous with respect to the Lebesgue measure. (b) The loss function $\phi : \mathbb{R} \to [0, \infty)$ is classification-calibrated, convex, non-increasing and continuously differentiable on its support.

Assumption 4.3 (a) and (b) are regular conditions ensuring that the target function $f_0$ is continuous and the loss $\phi$ is Lipschitz on bounded intervals. The absolute continuity assumption of the probability measure of $X$ with respect to the Lebesgue measure is reasonable for the approximate low-dimensional manifold case but incompatible with the exact low-dimensional manifold condition. This assumption is for deriving better $L_p$ approximation error bound, without which the error bound in term of the $L_\infty$ norm can still be established, but at the price of a wider neural network and a larger prefactor. It can be verified that commonly used loss functions, such as the cross entropy, the logistic and the exponential loss functions, satisfy Assumption 4.3 (b).

To deal with the approximation of unbounded target function, we truncate the target $f_0$ by a constant $T$, where $T$ may depend on $n$. Let $f_{0,T}$ be the truncated version of $f_0$ defined as

$$f_{0,T}(x) = \begin{cases} f_0(x), & \text{if } |f_0(x)| \leq T, \\ T\mathrm{sign}(f_0(x)), & \text{if } |f_0(x)| > T. \end{cases}$$

Denote the error of the loss function $\phi$ due to truncation by

$$\Delta_\phi(T) := \inf_{|a| \leq T} \phi(a) - \inf_{a \in \mathrm{Ran}(f_0)} \phi(a), \tag{17}$$

where $\mathrm{Ran}(f_0)$ is the range of the target function $f_0$. This error decreases as $T$ increases. Then, the approximation error can be decomposed into two terms that are easier to deal with.

**Lemma 4.4.** [$\phi$-approximation error] Suppose that Assumption 4.3 holds and $T \leq \mathcal{B}$. Then, the $\phi$-approximation error $\inf_{f \in \mathcal{F}_{\mathrm{CNN}}} R(f) - R(f_0)$ with respect to the loss function $\phi$ satisfies

$$\inf_{f \in \mathcal{F}_{\mathrm{CNN}}} R(f) - R(f_0) \leq B_\phi \inf_{f \in \mathcal{F}_{\mathrm{CNN}}} \mathbb{E}|f(X) - f_{0,T}(X)| + \Delta_\phi(T).$$

where $B_\phi$ is defined as the Lipschitz constant of $\phi$ on the interval $[-\mathcal{B}, \mathcal{B}]$.

We list $B_\phi$ and $\Delta_\phi(T)$ for hinge and cross entropy loss functions in Table 2. A detailed table including other loss functions are given in the appendix.

**Theorem 4.5.** [Approximation error bound] Suppose that Assumption 4.3 holds and $f_0 \in \mathcal{H}^\beta([0, 1]^d, B_0)$. For any $M, N \in \mathbb{N}^+$, let $\mathcal{F}_{\mathrm{CNN}}$ be the class of CNNs defined in (2) with $T \leq \mathcal{B}$, depth $L$ and filter lengths specified in (4) and (5). Then, the approximation error defined in (16) satisfies AppErr $\leq C(d, \beta)(NM)^{-2\beta/d} + \Delta_\phi(T)$, where $C(d, \beta) = 18B_\phi B_0(\beta + 1)^2 d^{\beta + (\beta \vee 1)/2}$.

There is a trade-off since $T \leq \mathcal{B}$ and $\Delta_\phi(T)$ deceases in $T$ but $C(d, \beta)$ increases in $B_\phi$ (or $\mathcal{B}$).

## 4.3 Non-asymptotic excess risk bound

Combining Theorems 4.2 and 4.5, we obtain the excess error bound.

**Theorem 4.6.** *[Non-asymptotic excess $\phi$-risk bound] Suppose that Assumption 4.3 holds and $f_0 \in \mathcal{H}^\beta([0,1]^d, B_0)$. For any $M, N \in \mathbb{N}^+$, let $\mathcal{F}_{CNN}$ be the class of CNNs defined in (2) with $T \leq \mathcal{B}$, depth $L$ and filter lengths specified in (4) and (5). Then, for any $\delta \in (0,1)$, with probability at least $1 - \delta$, the ERM $\hat{f}_n$ defined in (13) satisfies*

$$R(\hat{f}_n) - R(f_0) \leq StoErr + AppErr, \tag{18}$$

*with*

$$StoErr = \frac{2\phi_B}{\sqrt{n}}\left(C_0\sqrt{\mathcal{S}L\log(\mathcal{S})\log(n)} + \sqrt{\log(1/\delta)}\right), \; AppErr = C(d,\beta)(NM)^{-2\beta/d} + \Delta_\phi(T),$$

*where $C(d,\beta) = 18B_\phi B_0(\beta+1)^2 d^{\beta+(\beta\vee1)/2}$ and $C_0 > 0$ is a universal constant.*

The upper bound of the excess $\phi$-risk $R(\hat{f}_n) - R(f_0)$ in Theorem 4.6 is a sum of two error terms, the stochastic error and the approximation error. To achieve the optimal error rate, we need to balance the trade-off between them. On one hand, the bound for the stochastic error *StoError* increases with the size and the depth of $\mathcal{F}_{CNN}$. On the other hand, the bound for the approximation error *AppError* decreases with the depth and the upper bound of filter lengths of $\mathcal{F}_{CNN}$.

## 4.4 Circumventing the curse of dimensionality

We state a theorem that provides a non-asymptotic excess risk bound under the approximate low-dimensional manifold assumption.

**Theorem 4.7** (Circumventing the curse of dimensionality). *Suppose that Assumptions 2.3, 4.3 hold, and $f_0 \in \mathcal{H}^\beta([0,1]^d, B_0)$. For any $M, N \in \mathbb{N}^+$, let $\mathcal{F}_{CNN}$ be the class of CNNs defined in (2) with $T \leq \mathcal{B}$, depth $L$ and filter lengths and size $\mathcal{S}$ specified in (4), (5) and (6) with $\mathcal{W} = 38^2(\beta+1)^4 d_\varepsilon^{2\beta+2} N^2 \lceil\log_2(8N)\rceil^2$. Suppose $\rho$ in Assumption 2.3 satisfies $\rho \leq \rho_\varepsilon$, where $\rho_\varepsilon$ is defined in (10). Then for any $\delta \in (0,1)$, with probability at least $1 - \delta$, the ERM $\hat{f}_n$ defined in (13) satisfies*

$$R(\hat{f}_n) - R(f_0) \leq StoErr_* + AppErr_*, \tag{19}$$

*with*

$$StoErr_* = \frac{2\phi_B}{\sqrt{n}}\left(C_1\sqrt{\mathcal{S}L\log(\mathcal{S})\log(n)} + \sqrt{\log(1/\delta)}\right),$$

$$AppErr_* = \left((18 + C_2)\frac{B_\phi B_0}{(1-\varepsilon)^\beta}(\beta+1)^2 d^{1/2} d_\varepsilon^{3\beta/2}\right) \times (NM)^{-2\beta/d_\varepsilon} + \Delta_\phi(T),$$

*where $C_1, C_2 > 0$ are universal constants.*

For a high-dimensional input $X$ with a large $d$, $d_\varepsilon$ satisfies $d_\mathcal{M} \leq d_\varepsilon < d$ for $\varepsilon \in (0,1)$. For the stochastic error $StoErr_*$, comparing with the bound (18) in Theorem 4.6, we see that $StoErr_* = StoErr$, that is, the stochastic error does not change under the approximate low-dimensional manifold assumption. For the approximation error $AppErr_*$, we see that the convergence rate $(NM)^{-2\beta/d_\varepsilon}$ only depends on $d_\varepsilon$, which leads to a much faster convergence rate.

## 5 Examples

In this section, we illustrate the applications of Theorems 4.6 and 4.7 to obtaining non-asymptotic error bounds for the excess risk in classification. We apply the general excess risk bounds established in these theorems to several important classification methods with CNNs when a specific form of $\phi$ is given. We present the results for the cross entropy loss below. The non-asymptotic error bounds for the hinge, the logistic, the exponential and the least squares losses are given in Appendix **??**.

For the cross entropy loss $\phi(a) = -\log(0.5 + a)$, the Lipschitz constant $B_\phi$ on $[-0.5, 0.5]$ is not bounded. As the minimizer of $\phi$-risk $\eta - 0.5$ is bounded, we can choose $T = \mathcal{B} = 0.5 - n^{\beta/(2d+4\beta)}$. Then $\Delta_\phi(T) \leq n^{-\beta/(2d+4\beta)}$ and $B_\phi = n^{\beta/(2d+4\beta)}$.

Denote $\xi_n = n^{d/(2d+4\beta)}$. Take $N = 1$ in (7) and $M = \lfloor \xi \rfloor$ in (4) and let $\mathcal{F}_{\mathrm{CNN}}$ be the class of CNNs in (2) with depth $L \leq 378 \cdot 38^2 (\lfloor \beta \rfloor + 1)^6 d^{2\lfloor \beta \rfloor + 2} \lfloor \xi_n \rfloor \lceil \log_2(8\lfloor \xi_n \rfloor) \rceil / (s_{\min} - 1)$, filter lengths $2 \leq s_{\min} \leq s_{\max} \leq 9 \times 38^2 (\lfloor \beta \rfloor + 1)^4 d^{2\lfloor \beta \rfloor + 2}$ and size $\mathcal{S} \leq 8\mathcal{W}L \leq 42 * 8 * 9^2 * 38^4 (\lfloor \beta \rfloor + 1)^{10} d^{4\lfloor \beta \rfloor + 4} \lfloor \xi_n \rfloor \lceil \log_2(8\lfloor \xi_n \rfloor) \rceil / (s_{\min} - 1)$. Theorem 4.6 implies that, with probability at least $1 - \exp(-\xi_n^2)$, the excess $\phi$-risk of the ERM $\hat{f}_n$ defined in (13) satisfies

$$R(\hat{f}_n) - R(f_0) \leq Cn^{-\beta/(2d+4\beta)}(\log n)^2,$$

where $C = O(B_0(\beta + 1)^8 d^{3\beta+3}/(s_{\min} - 1))$ is a constant independent of $n$.

Let $T = \mathcal{B} = 0.5 - (\log n)^{-1}$. Using a modified cross entropy loss $\phi(a) = \max\{-\log(0.5 + a), \tau\}$ with $\tau = -\log(1 - (\log n)^{-1})$, the excess $\phi$-risk of the ERM $\hat{f}_n$ can be improved to $O(n^{-\beta/(d+2\beta)}(\log n)^2)$.

If the approximate low-dimensional manifold Assumption 2.3 holds and for any $\varepsilon \in (0, 1)$, the radius of the neighborhood $\rho$ in Assumption 2.3 satisfies $\rho \leq \rho_\varepsilon$, where $\rho_\varepsilon$ is defined in (10), then Theorem 4.7 implies that the rate of convergence can be improved:

$$R(\hat{f}_n) - R(f_0) \leq Cn^{-\beta/(2d_\varepsilon + 4\beta)}(\log n)^2,$$

where $C$ is a constant independent of $n$.

The above discussion shows that deep binary classifications with CNNs are adaptive to the low dimensional structure of the data and the smoothness of the target. Moreover, the prefactor $C$ depends on the ambient dimension $d$ polynomially, which improves the prefactors depending on $d$ exponentially in the existing excess risk bounds [46, 25, 30, 26, 45, 32, 33, 38]. Moreover, the filter lengths here are more flexible comparing with those requiring certain filter lengths in [32, 33, 48].

# 6   Conclusion

In this work, we derive new approximation error bounds with explicit prefactor in terms of network parameters for Sobolev-regular functions and Hölder smooth functions using deep convolutional networks. New approximation result when the target function is supported on an approximate lower-dimensional manifold is established. Different from existing results, the prefactor of our error bound depends on the ambient dimension polynomially instead of exponentially for Hölder smooth functions. The new approximation results are applied to establish non-asymptotic excess risk bounds for a class of classification methods using CNNs.

An important limitation of our work is that we only considered binary classification problems in applying our approximation results using deep convolutional networks. It would be interesting to apply the results in this work to other settings involving CNN approximation, such as the multiclass classification and regression problems. In addition, our work only partially explains the empirical successes of CNNs in practice. For image data, the assumption of approximate lower-dimensional support does not capture all the structural information. For example, spatial invariance is likely to be expected in some problems such as image classification. We believe that if such properties are taken into account, the theoretical bounds can be further improved. We hope to study this in the future.

## Acknowledgements

The authors wish to thank three anonymous reviewers for their comments and suggestions that helped improve the paper significantly.

G. Shen is partially supported by the research grant P0041243 from The Hong Kong Polytechnic University. Y. Jiao is supported in part by the National Science Foundation of China under Grant 11871474, the research fund of KLATASDSMOE, and the Fundamental Research Funds for the Central Universities NO. 2042022kf0071. Y. Lin is supported by the Hong Kong Research Grants Council (Grant Nos. 14306219 and 14306620), the National Natural Science Foundation of China (Grant No. 11961028) and Direct Grants for Research, The Chinese University of Hong Kong. J. Huang is partially supported by the research grant P0042888 from The Hong Kong Polytechnic University.

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
