# Supplementary to "Approximation with CNNs in Sobolev Space: with Applications to Classification"

**Guohao Shen**[*]
Department of Applied Mathematics, The Hong Kong Polytechnic University
Hung Hom, Kowloon, Hong Kong SAR, China
guohao.shen@polyu.edu.hk

**Yuling Jiao**[*]
School of Mathematics and Statistics
and Hubei Key Laboratory of Computational Science
Wuhan University, Wuhan 430072, China
yulingjiaomath@whu.edu.cn

**Yuanyuan Lin**[†]
Department of Statistics, The Chinese University of Hong Kong
Shatin, New Territories, Hong Kong SAR, China
ylin@sta.cuhk.edu.hk

**Jian Huang**[†]
Department of Applied Mathematics, The Hong Kong Polytechnic University
Hung Hom, Kowloon, Hong Kong SAR, China
j.huang@polyu.edu.hk

## Supplementary materials

In the Supplementary materials, we include detailed descriptions on convex surrogate losses, convolutional neural networks, non-asymptotic error bounds for commonly used loss functions, and prove Theorems 2.1, 2.2, 2.4, 4.4, 4.6, and 4.7, as well as the error bounds for cross entropy, SVM, logistic, exponential and least squares loss examples. A toy example on the numerical performance of CNN approximation is presented in Appendix D.

## 1 Convex surrogate loss

We next give a brief review of the convex surrogate loss functions and discuss in details on the connection between the excess risk with respect to the $\phi$-loss and that of 0-1 loss [28, 4].

Let $\phi$ be a given convex univariate function $\phi : \mathbb{R} \to [0, \infty)$. Instead of minimizing the excess risk $R$ over $\mathcal{H}$, we consider minimizing the risk with respect to the loss $\phi$ ($\phi$-risk)

$$R(f) := \mathbb{E}\{\phi(Yf(X))\}$$

over a certain class of functions $\mathcal{F}$, where $\phi : \mathbb{R} \to [0, \infty)$ is some generic loss function. For the special case when $\mathcal{H} = \{h : h(x) = \text{sign}(f(x)), f \in \mathcal{F}\}$ and $\phi(\cdot)$ is a step function, i.e., $\phi(x) = 1$

---

[*]Guohao Shen and Yuling Jiao contributed equally to this work
[†]Corresponding authors

36th Conference on Neural Information Processing Systems (NeurIPS 2022).

if $x < 0$ and $\phi(x) = 0$ if $x \geq 0$, then to minimize $R$ over $\mathcal{H}$, it suffices to minimize $R$ over $\mathcal{F}$. Given $S = \{(X_i, Y_i)\}_{i=1}^n$, let $R_n(f) := \sum_{i=1}^n \phi(Y_i f(X_i))/n$ be the empirical risk of $f$ w.r.t the $\phi$-loss, and define the empirical risk estimator (ERM) as

$$\hat{f}_n \in \arg\min_{f \in \mathcal{F}} R_n(f). \tag{1}$$

Based on the ERM $\hat{f}_n$, a classification rule or a classifier $\hat{h}_n(x) := \text{sign}(\hat{f}_n(x))$ for $x \in \mathcal{X}$ can be induced which aims at minimizing the 0-1 risk. As shown in [28] and [4], for a properly chosen $\phi$, $\hat{f}_n$ can indeed help reduce the 0-1 excess risk $R_*(\hat{h}_n) - R_*(h_0)$. More precisely, let $R_0 := \inf_{f \text{ measurable}} R(f)$, then for a proper $\phi$, we have

$$\psi(R_*(\hat{h}_n) - R_*(h_0)) \leq R(\hat{f}_n) - R(f_0),$$

where $\psi : [-1, 1] \to [0, \infty)$ is a nonnegative continuous function, invertible on $[0, 1]$, and achieves its minimum at 0 with $\psi(0) = 0$. A wide variety of popular classification methods are based on this tactic. For instance, when $\phi(a) = \log\{1 + \exp(-a)\}$, it is the logistic regression [11]; when $\phi(a) = \max\{1 - a, 0\}$, it becomes the SVM [7]; when $\phi(a) = \exp(-a)$ is the exponential loss, it is the AdaBoost algorithm [10]; when $\phi(a) = (1 - a)^2$, it is the least squares method for classification, and so on.

Define the measurable minimizer of $R$ as

$$f_0 = \arg\min_{f \text{ measurable}} \mathbb{E}\{\phi(Yf(X))\}, \tag{2}$$

and the corresponding minimal $\phi$-risk as $R_0 = R(f_0)$. Clearly, the optimal $f_0$ depends on the loss function $\phi$. If $\phi$ is not properly chosen, the resulting $f_0$ can be poor and thus the classifier $h_0(x) = \text{sign}(f_0(x))$ based on $f_0$ can be invalid. To study the basic conditions imposed on $\phi$, so as to produce a valid classifier towards minimizing the 0-1 loss, we need a thorough understanding of the risk $R$.

Let the conditional $\phi$-risk of $f$ given $X = x$ be denoted by $R_x(f) := \mathbb{E}\{\phi(Yf(X)) \mid X = x\}$. Recall that $\eta(x) = \mathbb{P}\{Y = 1 \mid X = x\}$ as defined in (15). We have

$$R_x(f) = \eta(x)\phi(f(x)) + \{1 - \eta(x)\}\phi(-f(x)), \ x \in \mathcal{X}.$$

For a good classifier, if $\eta(x) > 1/2$, it is naturally expected that $f_0(x) > 0$ and $h_0(x) = \text{sign}(f_0(x)) = 1$ (correct sign); thus to encourage $f_0(x) > 0$, we should at least require $\phi(f_0(x)) < \phi(-f_0(x))$ to minimize the conditional risk $R_x$. Similarly, if $\eta(x) < 1/2$, we expect that the contrary happens. A rigorous definition for such an ideal $\phi$ is given in Definition 1 of [4].

**Definition 1.1** (Classification-calibrated). For $\eta \in [0, 1]$ and $a \in \mathbb{R}$, define $H(\eta, a) = \eta\phi(a) + (1 - \eta)\phi(-a)$. Then, we say that $\phi$ is *classification-calibrated* if for any $\eta \neq 1/2$,

$$\inf_{a \in \mathbb{R}} H(\eta, a) < \inf_{a \in \mathbb{R}: a(2\eta - 1) \leq 0} H(\eta, a).$$

With a classification-calibrated loss $\phi$, it is guaranteed that the "incorrect" label $\text{sign}(f(X))$ in the sense that it is inconsistent with the Bayes estimator $\text{sign}(2\eta(X) - 1))$ results in a strictly larger loss under $\phi$. It was shown by [4] that, the surrogate loss function $\phi$ is able to produce the optimal Bayes classifier. For ease of reference, we state this result as Lemma 1.2 below.

**Lemma 1.2** (Theorem 1 of [4]). *For any nonnegative loss function $\phi : \mathbb{R} \to [0, \infty)$, any measurable $f : \mathcal{X} \to \mathbb{R}$, its induced classifier $h_f = \text{sign}(f) : \mathcal{X} \to \{\pm 1\}$, and any probability measure $\mathbb{P}$ of $(X, Y)$ on $\mathcal{X} \times \{\pm 1\}$,*

$$\psi(R_*(h_f) - R_*(h_0)) \leq R(f) - R_0,$$

*where $\psi : [-1, 1] \to [0, \infty)$ is the Fenchel-Legendre biconjugate of $\tilde{\psi} : [-1, 1] \to \mathbb{R}$, and*

$$\tilde{\psi}(\theta) = \inf_{a \in \mathbb{R}: a(2\eta - 1) \leq 0} H(\frac{1 + \theta}{2}) - \inf_{a \in \mathbb{R}} H(\frac{1 + \theta}{2}).$$

*Besides, if $\phi$ is classification-calibrated, then for any sequence $\{a_m\}$ in $[0, 1]$, $\psi(a_m) \to 0$, if and only if $a_m \to 0$.*

Lemma 1.2 has some important implications. First, for any nonnegative $\phi$, $\psi$ is simply the functional convex hull (the greatest convex minorant) of $\tilde{\psi}$. Both $\psi$ and $\tilde{\psi}$ are continuous on $[-1, 1]$, and $\psi$ is nonnegative that attains its minimum at 0 with $\psi(0) = 0$. Second, if the loss $\phi$ is nonnegative and classification-calibrated, then $\psi(\theta) > 0$ for all $\theta \in (0, 1]$ and $\psi$ is invertible on $[0, 1]$. In this case, we have $R_*(h_f) - R_*(h_0) \leq \psi^{-1}(R(f) - R_0)$. Third, if $\phi$ is nonnegative convex and classification-calibrated, then $\psi(\theta) = \phi(0) - \inf_{a \in \mathbb{R}} H((1 + \theta)/2, a)$, which gives an easy way to compute the function $\psi$. Note that for the formulation of $\psi$, Theorem 34 in [6] shows that if $\phi$ is convex, $\phi''(0) > 0$ exists and $\phi' < 0$, then $\psi(\theta) = \theta^2$.

Next, we present several examples of $\phi$, and the corresponding $f_0$, $R(f_0)$, $\psi$ and its inverse on $[0, 1]$ in Table 1.

Table 1: Minimizer and minimal conditional risk under different loss functions $\phi$.

|  | Least squares | SVM | Exponential | Logistic | Cross entropy |
|---|---|---|---|---|---|
| $\phi(a)$ | $(1-a)^2$ | $\max\{1-a, 0\}$ | $\exp(-a)$ | $\log\{1 + \exp(-a)\}$ | $-\log\{0.5 + a\}$ |
| $f_0(x)$ | $2\eta - 1$ | $\text{sign}(2\eta - 1)$ | $\frac{1}{2}\log(\frac{\eta}{1-\eta})$ | $\log(\frac{\eta}{1-\eta})$ | $\eta - 0.5$ |
| $R_x(f_0)$ | $4\eta(1-\eta)$ | $1 - |2\eta - 1|$ | $2\sqrt{\eta(1-\eta)}$ | $-\eta\log(\eta) - (1-\eta)\log(1-\eta)$ | $-\eta\log(\eta) - (1-\eta)\log(1-\eta)$ |
| $\psi(\theta)$ | $\theta^2$ | $|\theta|$ | $1 - \sqrt{1-\theta^2}$ | $\theta^2$ | $\theta^2$ |
| $\psi^{-1}(\theta)$ | $\sqrt{\theta}$ | $|\theta|$ | $\sqrt{1 - (1-\theta)^2}$ | $\sqrt{\theta}$ | $\sqrt{\theta}$ |

Note: $\eta(x)$ is written as $\eta$ for notational simplicity.

We also list the Lipschitz constant $B_\phi$ and $\Delta_\phi(T)$ for commonly-used SVM, cross entropy, least squares, exponential and logistic loss functions in Table 2.

Table 2: Bound $\phi_B$ of $\phi$, Lipschitz constant $B_\phi$ and $\Delta_\phi(T)$ for the truncated $f_0$ under least squares, logistic, exponential and SVM hinge loss function restricted to $[-\mathcal{B}, \mathcal{B}]$ for $1 \leq T \leq \mathcal{B}$; and cross entropy loss function restricted to $[-\mathcal{B}, \mathcal{B}]$ for $T \leq \mathcal{B} \leq 0.5$.

|  | Least squares | SVM | Exponential | Logistic | Cross entropy |
|---|---|---|---|---|---|
| $\phi(a)$ | $(1-a)^2$ | $\max\{1-a, 0\}$ | $\exp(-a)$ | $\log\{1 + \exp(-a)\}$ | $-\log\{0.5 + a\}$ |
| $f_0(x)$ | $2\eta - 1$ | $\text{sign}(2\eta - 1)$ | $\frac{1}{2}\log(\frac{\eta}{1-\eta})$ | $\log(\frac{\eta}{1-\eta})$ | $\eta - 0.5$ |
| $\phi_B$ | $(\mathcal{B}+1)^2$ | $1$ | $\exp(\mathcal{B})$ | $\log(1 + \exp(\mathcal{B}))$ | $-\log(0.5 - \mathcal{B})$ |
| $B_\phi$ | $2\mathcal{B} + 2$ | *** | $\exp(\mathcal{B})$ | $1/\{\exp(-\mathcal{B}) + 1\}$ | $1/(0.5 + \mathcal{B})$ |
| $\Delta_\phi(T)$ | $0$ | $0$ | $\exp(-T)$ | $\log\{1 + \exp(-T)\}$ | $-\log\{1 + (0.5 - T)\}$ |

Note: $\eta(x)$ is written as $\eta$ for notational simplicity and "***" stands for not applicable.

## 2 Convolutional neural networks

As indicated by their name, CNNs employ a mathematical operation called convolution. Convolutional networks are a specialized type of structured sparse feedforward neural network (FNN) that use convolution in place of general matrix multiplication in at least one of their layers [12]. There are different formulations of CNNs in the literature [1, 31, 22, 18, 29, 30].

For a general neural network, let $L$ denote the number of layers and $(\sigma_1, \ldots, \sigma_L)$ denote the activation functions, such as the rectified linear unit (ReLU) and max pooling function. Besides, let $d_i$ denote the width (the number of neurons or computational units) of the $i$-th layer, $i = 1, \ldots, L + 1$, and $\mathcal{W} = \max\{d_1, \ldots, d_L\}$ denote the maximum width among layers. In our present classification problem, $d_0 = d$ (the dimension of the input $X$) and $d_{L+1} = 1$ (the dimension of the output $Y$).

In the following, two common types of neural networks including FNNs or MLPs and CNNs with downsampling [29] are shown.

- The architecture of a MLP $f_{MLP} : \mathbb{R}^d \to \mathbb{R}$ can be expressed as a composition of a series of linear transformations: for any $x \in \mathbb{R}^d$,

$$f_{MLP}(x) = W_{L+1}\sigma_L(W_L\sigma_{L-1}(\cdots\sigma_2(W_2\sigma_1(W_1 x + b_1) + b_2)\cdots) + b_L) + b_{L+1}, \quad (3)$$

where $W_i \in \mathbb{R}^{d_{i+1} \times d_i}$ is a fully connected weight matrix, and $b_i \in \mathbb{R}^{d_{i+1}}$ is a bias vector in the $i$-th linear transformation. Usually, the activation functions $(\sigma_1, \ldots, \sigma_L)$ are the ReLU activation, where $\mathrm{ReLU}(x) = \max\{0, x\}$ for any $x \in \mathbb{R}$ (defined for each component of $x$ if $x$ is a vector).

- The formulation of a downsampled CNN $f_{\mathrm{CNN}} : \mathbb{R}^d \to \mathbb{R}$ is essentially a specially structured architectures of MLP, and here we use the formulation defined in [31, 29, 30]

$$f_{\mathrm{CNN}} = A_{L+1} \circ A_L \circ \cdots \circ A_2 \circ A_1, \tag{4}$$

where $\circ$ denotes the functional composition and $A_i$ is a linear operation, $i = 1, \ldots, L+1$. The $A_i$'s are either convolutional operators or downsampling operators. For convolutional layers,

$$A_i = \sigma(W_i^c x + b_i^c),$$

where $\sigma$ is the rectified linear unit (ReLU) activation function applying to each component of the input vector, $W_i^c \in \mathbb{R}^{d_i \times d_{i-1}}$ is the structured sparse Toeplitz type weight matrix induced by the convolutional filter $\{w^{(i)}_j\}_{j=0}^{s^{(i)}}$ with filter length $s^{(i)} \in \mathbb{N}$ involving only $s^{(i)} + 1$ free parameters. The corresponding $(d_{i-1} + s^{(i)}) \times d_{i-1}$ Toeplitz type weight matrix $W_i^c$ is given explicitly by

$$W_i^c = \begin{bmatrix} w_0^{(i)} & 0 & 0 & 0 & \cdots & \cdots & 0 \\ w_1^{(i)} & w_0^{(i)} & 0 & 0 & \cdots & \cdots & 0 \\ \vdots & \ddots & \ddots & \ddots & \ddots & \ddots & \vdots \\ w_{s^{(i)}}^{(i)} & w_{s^{(i)}-1}^{(i)} & \cdots & w_0^{(i)} & 0 & \cdots & 0 \\ 0 & w_{s^{(i)}}^{(i)} & \cdots & w_1^{(i)} & w_0^{(i)} & 0\cdots & 0 \\ \vdots & \ddots & \ddots & \ddots & \ddots & \ddots & \vdots \\ 0 & \cdots & 0 & w_{s^{(i)}}^{(i)} & \cdots & w_1^{(i)} & w_0^{(i)} \\ 0 & \cdots & 0 & 0 & w_{s^{(i)}}^{(i)} & \cdots & w_1^{(i)} \\ \vdots & \ddots & \ddots & \ddots & \ddots & \ddots & \vdots \\ 0 & 0 & 0 & 0 & \cdots & 0 & w_{s^{(i)}}^{(i)} \end{bmatrix} \in \mathbb{R}^{(d_{i-1}+s^{(i)}) \times d_{i-1}}.$$

It can be seen that by convolutional operations here, the network has linearly increasing widths $d_i = d_{i-1} + s^{(i)}$ for convolutional layers. On the contrary, the downsampling layers decrease the width of the networks.

For downsampling layers,

$$A_i(x) = D_i(x) = W_i^D x,$$

where $D_i : \mathbb{R}^{d_{i-1}} \to \mathbb{R}^{[d_{i-1}/m_i]}$ is the downsampling operator with a scaling parameter $m_i \leq d_{i-1}$ and $W^D \in \mathbb{R}^{d_{i-1} \times [d_{i-1}/m_i]}$ is corresponding structured sparse weight matrix induced by $D_i$ in the $i$-th layer. Here the downsampling operator [29] is given by

$$D_i(x) = (x_{jm_i})_{j=1}^{[d_{i-1}/m_i]}, \qquad x \in \mathbb{R}^{d_{i-1}},$$

which takes a real vector $x \in \mathbb{R}^{d_{i-1}}$ as input and outputs a subvector of $x$ with length $\mathbb{R}^{[d_{i-1}/m_i]}$ where $[a]$ denotes the integer part of $a \in \mathbb{R}$. It can be seen that by downsampling operation $D_i$ here, the network width is scaled down by about $m_i$ times. For the function class of the downsampled CNNs formulated as in (4), we let $L$ be th number of hidden layers and $\mathcal{S}$ be the total number of parameters for networks in $\mathcal{F}_{\mathrm{CNN}}$ and let $\mathcal{W}$ be the maximum filter length of convolutional layers.

A downsampled CNN $f_{CNN}$ is called uniform if the length of its filters $\{s^{(i)}\}$ in the convolutional layers are the same. [29] showed that any multilayer perceptron can be represented by a uniform downsampled CNN with parameters no more than 8 times as that of the MLP.

## 3 Non-asymptotic error bounds for commonly used loss functions

In this section, we show more examples of the non-asymptotic error bounds for commonly used loss functions including the hinge loss, the logistic loss, the exponential loss and the least square loss.

### 3.1 SVM: the hinge loss

For SVM, the loss function $\phi(a) = \max\{1-a, 0\}$ is not differentiable at $a = 1$ and thus the $\phi$-risk minimizer $f_0(x) = \text{sign}(2\eta(x) - 1)$ may not be continuous even though $\eta$ is continuous, as $f_0(x)$ is discontinuous when $\eta(x) = 1/2$. To tackle this problem, we additionally impose the low noise condition on $\eta$ [20, 24], i.e., there exist $c_{\text{noise}} > 0$ and $q \in [0, \infty]$ such that for any $t > 0$, $\mathbb{P}(|2\eta(X) - 1| \leq t) \leq c_{\text{noise}} t^q$, where the constant $q$ is called *the noise exponent*.

Suppose that Assumptions 4.3 holds $f_0 \in \mathcal{H}^\beta([0,1]^d, B_0)$. Denote $\zeta_n = n^{d/\{2d + 4\beta(q+1)\}}$. Let $\mathcal{F}_{\text{CNN}}$ be the class of CNNs defined in (2) with $\mathcal{B} \geq 1$, depth $L \leq 378 \cdot 38^2(\lfloor\beta\rfloor + 1)^6 d^{2\lfloor\beta\rfloor+2} \lfloor \zeta_n \rfloor \lceil \log(8\zeta_n) \rceil / (s_{\min} - 1)$, filter lengths $2 \leq s_{\min} \leq s_{\max} \leq 9 \times 38^2(\lfloor\beta\rfloor+1)^4 d^{2\lfloor\beta\rfloor+2}$ and size $\mathcal{S} \leq 8\mathcal{W}L \leq 42*8*9^2*38^4(\lfloor\beta\rfloor+1)^{10} d^{4\lfloor\beta\rfloor+4} \lfloor \zeta_n \rfloor \lceil \log(8\zeta_n) \rceil / (s_{\min} - 1)$. Theorem 4.6 implies that, with probability at least $1 - \exp(-\zeta_n^2)$, the excess $\phi$-risk of the ERM $\hat{f}_n$ defined in (13) satisfies

$$R(\hat{f}_n) - R(f_0) \leq Cn^{-\beta(q+1)/\{d+2\beta(q+1)\}}(\log n)^2,$$

where $C = O(c_{\text{noise}} 4^q B_0^{q+1}(\beta+1)^{2q+8} d^{(q+1)(3\beta+3)})$. For the induced classifier $\hat{h}_n = \text{sign}(\hat{f}_n)$, the excess misclassification error satisfies

$$R_*(\hat{h}_n) - R_*(h_0) \leq Cn^{-\beta(q+1)/\{d+2\beta(q+1)\}}(\log n)^2.$$

The excess risk bound depends on $q$, the noise exponent. When $q = 0$ (high noise), the convergence rate is $n^{-\beta/(d+2\beta)}$; when $q = +\infty$ (no noise), the rate will be significantly improved to $n^{-1/2}$. A similar result can be found in Theorem 3.3 of [16].

Suppose Assumption 2.3 holds and for any $\varepsilon \in (0, 1)$, the radius of the neighborhood $\rho$ in this assumption satisfies $\rho \leq \rho_\varepsilon$, then Theorem 4.7 implies that

$$R(\hat{f}_n) - R(f_0) \leq Cn^{-\beta(q+1)/\{0.75d_\varepsilon + 2\beta(q+1)\}}(\log n)^2,$$

where $C$ is a constant independent of $n$.

### 3.2 The logistic loss

For the logistic loss function $\phi(a) = \log\{1 + \exp(-a)\}$, the minimizer of $\phi$-risk $\log\{\eta(x)/1 - \eta(x)\}$ is unbounded with the Lipschitz constant $B_\phi = 1/\{\exp(-\mathcal{B}) + 1\} \leq 1$ on $[-\mathcal{B}, \mathcal{B}]$, and $\phi_B = \log(1 + \exp(\mathcal{B})) \leq \mathcal{B} + \infty$ and the truncation error $\Delta_\phi(T) = \log\{1 + \exp(-T)\}$. A feasible choice is $T = \mathcal{B} = \log(n)$ where $n$ is the sample size, in which case $\Delta_\phi(T) = \log\{1 + 1/n\} \leq 1/n$.

Suppose Assumptions 4.3 (in main context) holds and $f_0 \in \mathcal{H}^\beta([0,1]^d, B_0)$. Take $N = 1$ in (7) and $M = \lfloor \xi \rfloor$ in (4) and let $\mathcal{F}_{\text{CNN}}$ be the class of CNNs in (2) with depth $L \leq 378 \cdot 38^2(\lfloor\beta\rfloor + 1)^6 d^{2\lfloor\beta\rfloor+2} \lfloor n^{d/\{2d+4\beta\}} \rfloor \lceil \log(8n) \rceil / (s_{\min} - 1)$, filter lengths $2 \leq s_{\min} \leq s_{\max} \leq 9 \times 38^2(\lfloor\beta\rfloor + 1)^4 d^{2\lfloor\beta\rfloor+2}$ and size $\mathcal{S} \leq 8\mathcal{W}L \leq 42*8*9^2*38^4(\lfloor\beta\rfloor+1)^{10} d^{4\lfloor\beta\rfloor+4} \lfloor n^{d/\{2d+4\beta\}} \rfloor \lceil \log(8n) \rceil / (s_{\min} - 1)$.

Theorem 4.6 implies that, with probability at least $1 - \exp\{-n^{d/(d+2\beta)}\}$, the excess $\phi$-risk of the ERM $\hat{f}_n$ defined in (13) satisfies

$$R(\hat{f}_n) - R(f_0) \leq n^{-1} + (\log n) n^{-\beta/(d+2\beta)}$$
$$\times \{c(\lfloor\beta\rfloor + 1)^8 d^{3\lfloor\beta\rfloor+3}(\log n)^2/(s_{\min} - 1) + 18 B_0(\lfloor\beta\rfloor + 1)^2 d^{\lfloor\beta\rfloor+(\lfloor\beta\rfloor\vee 1)/2}\},$$

or simply

$$R(\hat{f}_n) - R(f_0) \leq C(d, \beta, B_0, s_{\min})(\log n)^3 n^{-\beta/(d+2\beta)},$$

where $C(d, \beta, B_0, s_{\min}) = O(B_0(\lfloor\beta\rfloor+1)^8 d^{3\lfloor\beta\rfloor+3}/(s_{\min} - 1))$ is a constant independent of $n$. For the induced classifier $\hat{h}_n = \text{sign}(\hat{f}_n)$, the excess misclassification error satisfies

$$R_*(\hat{h}_n) - R_*(h_0) \leq \sqrt{C(d, \beta, B_0, s_{\min})}(\log n)^{3/2} n^{-\beta/(2d+4\beta)}.$$

In addition, suppose the approximate low-dimensional manifold Assumption 2.3 also holds and for any $\varepsilon \in (0, 1)$, the radius of the neighborhood $\rho$ in Assumption 2.3 satisfies $\rho \leq$

$C_2(NM)^{-2\beta/d_\varepsilon}(\lfloor\beta\rfloor+1)^2 d^{1/2}d_\varepsilon^{3\lfloor\beta\rfloor}(\sqrt{d/d_\varepsilon}+1-\varepsilon)^{-1}(1-\varepsilon)^{1-\beta}$, where $d_\varepsilon = O(d_\mathcal{M}\log(d/\varepsilon)/\varepsilon^2)$ is an integer satisfying $d_\mathcal{M}\leq d_\varepsilon \ll d$. Then Theorem 4.7 shows that the rate of convergence can be improved to

$$R(\hat{f}_n) - R(f_0) \leq C(d, d_\varepsilon, \beta, B_0, s_{\min})(\log n)^2 n^{-\beta/(d_\varepsilon+2\beta)},$$

where $C(d, d_\varepsilon, \varepsilon, \beta, B_0, s_{\min}) = O(B_0(1-\varepsilon)^{-\beta}(\lfloor\beta\rfloor+1)^8 d^{1/2}d_\varepsilon^{3\lfloor\beta\rfloor+3}/(s_{\min}-1))$ is a constant independent of $n$.

### 3.3 The exponential loss

For the exponential loss function $\phi(a) = \exp(-a)$, the minimizer of $\phi$-risk $\log\{\eta(x)/1 - \eta(x)\}/2$ is unbounded, the truncation error $\Delta_\phi(T) = \exp(-T)$, the Lipschitz constant $B_\phi = \exp(\mathcal{B})$ and $\phi_B = \exp(\mathcal{B})$. A feasible choice is $T = \mathcal{B} = (\beta/\{2d+4\beta\})\log(n)$ where $n$ is the sample size.

Suppose Assumptions 4.3 (in main context) holds and $f_0 \in \mathcal{H}^\beta([0,1]^d, B_0)$. Take $N = 1$ in (7) and $M = \lfloor\xi\rfloor$ in (4) and let $\mathcal{F}_{\text{CNN}}$ be the class of CNNs in (2) with depth $L \leq 378\cdot38^2(\lfloor\beta\rfloor+1)^6 d^{2\lfloor\beta\rfloor+2}\lfloor n^{d/\{2d+4\beta\}}\rfloor\lceil\log(8n)\rceil/(s_{\min}-1)$, filter lengths $2 \leq s_{\min} \leq s_{\max} \leq 9\times38^2(\lfloor\beta\rfloor+1)^4 d^{2\lfloor\beta\rfloor+2}$ and size $\mathcal{S} \leq 8WL \leq 42*8*9^2*38^4(\lfloor\beta\rfloor+1)^{10} d^{4\lfloor\beta\rfloor+4}\lfloor n^{d/\{2d+4\beta\}}\rfloor\lceil\log(8n)\rceil/(s_{\min}-1)$. Theorem 4.6 implies that, with probability at least $1 - \exp\{-n^{d/(d+2\beta)}\}$, the excess $\phi$-risk of the ERM $\hat{f}_n$ defined in (13) satisfies

$$R(\hat{f}_n) - R(f_0) \leq n^{-\beta/(2d+4\beta)} + n^{-\beta/(2d+4\beta)}$$
$$\times\{c(\lfloor\beta\rfloor+1)^8 d^{3\lfloor\beta\rfloor+3}(\log n)^2/(s_{\min}-1) + 18B_0(\lfloor\beta\rfloor+1)^2 d^{\lfloor\beta\rfloor+(\lfloor\beta\rfloor\vee1)/2}\},$$

or simply

$$R(\hat{f}_n) - R(f_0) \leq C(d, \beta, B_0, s_{\min})(\log n)^2 n^{-\beta/(2d+4\beta)},$$

where $C(d, \beta, B_0, s_{\min}) = O(B_0(\lfloor\beta\rfloor+1)^8 d^{3\lfloor\beta\rfloor+3}/(s_{\min}-1))$ is a constant independent of $n$.

In addition, suppose the approximate low-dimensional manifold Assumption 2.3 also holds and for any $\varepsilon \in (0,1)$, the radius of the neighborhood $\rho$ in Assumption 2.3 satisfies $\rho \leq C_2(NM)^{-2\beta/d_\varepsilon}(\lfloor\beta\rfloor+1)^2 d^{1/2}d_\varepsilon^{2\lfloor\beta\rfloor}(\sqrt{d/d_\varepsilon}+1-\varepsilon)^{-1}(1-\varepsilon)^{1-\beta}$, where $d_\varepsilon = O(d_\mathcal{M}\log(d/\varepsilon)/\varepsilon^2)$ is an integer satisfying $d_\mathcal{M}\leq d_\varepsilon \ll d$. Then Theorem 4.7 shows that the rate of convergence can be improved to

$$R(\hat{f}_n) - R(f_0) \leq C(d, d_\varepsilon, \varepsilon, \beta, B_0, s_{\min})(\log n)^2 n^{-\beta/(2d_\varepsilon+4\beta)},$$

where $C(d, d_\varepsilon, \varepsilon, \beta, B_0, s_{\min}) = O(B_0(1-\varepsilon)^{-\beta}(\lfloor\beta\rfloor+1)^8 d^{1/2}d_\varepsilon^{3\lfloor\beta\rfloor+3}/(s_{\min}-1))$ is a constant independent of $n$.

To improve the rate of convergence, one can use a modified exponential loss function, i.e. $\phi(a) = \max\{\exp(-a), \tau\}$ for some small $\tau > 0$. The minimum of modified $\phi$ can be achieved at $a^*(\tau) = -\log(\tau)$, and the minimizer $f_0(x)$ will be a truncated version of $\log(\eta/(1-\eta))/2$ (truncated by $a^*(\tau) = -\log(\tau)$). In light of this, under the modified exponential loss with $\tau = \exp(-T)$, the corresponding measurable minimizer defined in (14) is $f_{0,T}$, the truncated version of the minimizer $f_0$ under the original exponential loss; see Table 2. Then $\Delta_\phi(T) = 0$ by its definition since the infimum of the modified loss can be achieved within $[-T, T]$. By choosing $T = \mathcal{B} = \log\log n$ and $\tau = \exp(-T)$, with the modified exponential loss, Theorem 4.6 implies that with probability at least $1 - \exp\{-n^{d/(d+2\beta)}\}$, the excess $\phi$-risk of the ERM $\hat{f}_n$ defined in (13) satisfies

$$R(\hat{f}_n) - R(f_0) \leq C(d, \beta, B_0, s_{\min})(\log n)^3 n^{-\beta/(d+2\beta)},$$

where $C(d, \beta, B_0, s_{\min}) = O(B_0(\lfloor\beta\rfloor+1)^8 d^{3\lfloor\beta\rfloor+3}/(s_{\min}-1))$ is a constant independent of $n$. For the induced classifier $\hat{h}_n = \text{sign}(\hat{f}_n)$, the excess misclassification error satisfies

$$R_*(\hat{h}_n) - R_*(h_0) \leq \sqrt{C(d, \beta, B_0, s_{\min})}(\log n)^{3/2} n^{-\beta/(2d+4\beta)}.$$

## 3.4 The least squares loss

For the least squares loss $\phi(a) = (1-a)^2$, we first prove that the approximation error is of a special form. For any $f \in \mathcal{F}_{\mathrm{CNN}}$

$$
\begin{aligned}
R(f) - R(f_0) &= \mathbb{E}\{\phi(Yf(X)) - \phi(Yf_0(X))\} \\
&= \mathbb{E}\{(1 - Yf(X))^2 - (1 - Yf_0(X))^2\} \\
&= \mathbb{E}[f(X)^2 - f_0(X)^2 - 2Y\{f(X) - f_0(X)\}] \\
&= \mathbb{E}[(f(X) - f_0(X))\{f(X) + f_0(X) - 2\mathbb{E}(Y \mid X)\}] \\
&= \mathbb{E}[(f(X) - f_0(X))(f(X) + f_0(X) - 2f_0(X))] \\
&= \mathbb{E}|f(X) - f_0(X)|^2.
\end{aligned}
$$

Ans we have

$$
\inf_{f \in \mathcal{F}_{\mathrm{CNN}}} R(f) - R(f_0) = \inf_{f \in \mathcal{F}_{\mathrm{CNN}}} \mathbb{E}|f(X) - f_0(X)|^2.
$$

Besides, note that for least squares $f_0 = 2\eta - 1$ and $\|f_0\|_\infty = 1$, thus taking the truncation $T = 1$ is sufficient for the error control analysis. Let $\mathcal{F}_{\mathrm{CNN}}$ be a class of CNNs defined in (2) with $\mathcal{B} \geq T = 1$.

By Theorem 2.2, there exists a function $f$ implemented by CNN with depth $L \leq 378 \cdot 38^2(\lfloor \beta \rfloor + 1)^6 d^{2\lfloor \beta \rfloor + 2} M \lceil \log(8M) \rceil / (s_{\min} - 1)$, filter lengths $2 \leq s_{\min} \leq s_{\max} \leq 9 \times 38^2(\lfloor \beta \rfloor + 1)^4 d^{2\lfloor \beta \rfloor + 2}$ and size $\mathcal{S} \leq 8\mathcal{W}L \leq 42 * 8 * 9^2 * 38^4(\lfloor \beta \rfloor + 1)^{10} d^{4\lfloor \beta \rfloor + 4} N \lceil \log(8N) \rceil / (s_{\min} - 1)$ such that

$$
\mathbb{E}\|f(X) - f_0(X)\|_2^2 \leq 18^2 B_0^2 (\lfloor \beta \rfloor + 1)^4 d^{2\lfloor \beta \rfloor + (\beta \vee 1)} (NM)^{-4\beta/d},
$$

and based on Lemma 4.1 and Theorem 4.2, the empirical $\phi$-risk minimizer $\hat{f}_n$ defined in (13) satisfies, for any $\delta > 0$, with probability at least $1 - \delta$,

$$
\begin{aligned}
R(\hat{f}_n) - R(f_0) \leq {} & \frac{\phi_B}{\sqrt{n}}\left(\sqrt{c\mathcal{S}L \log(\mathcal{S})\log n} + \sqrt{\log(1/\delta)}\right) \\
& + 18^2 B_0^2(\lfloor \beta \rfloor + 1)^4 d^{2\lfloor \beta \rfloor + (\beta \vee 1)}(NM)^{-4\beta/d}.
\end{aligned}
$$

Let $\mathcal{F}_{\mathrm{CNN}}$ be the class of CNNs defined in (2) with depth $L \leq 378 \cdot 38^2(\lfloor \beta \rfloor + 1)^6 d^{2\lfloor \beta \rfloor + 2}\lfloor n^{d/(2d+8\beta)}\rfloor\lceil \log(8n^{d/(2d+8\beta)})\rceil / (s_{\min} - 1)$, filter lengths $2 \leq s_{\min} \leq s_{\max} \leq 9 \times 38^2(\lfloor \beta \rfloor + 1)^4 d^{2\lfloor \beta \rfloor + 2}$ and size $\mathcal{S} \leq 8\mathcal{W}L \leq 42 * 8 * 9^2 * 38^4(\lfloor \beta \rfloor + 1)^{10} d^{4\lfloor \beta \rfloor + 4}\lfloor n^{d/(2d+8\beta)}\rfloor\lceil \log(8n^{d/(2d+8\beta)})\rceil / (s_{\min} - 1)$, $\mathcal{B} \geq B_0$ and $\delta = \exp\{-n^{d/(d+4\beta)}\}$. By plunging in $\phi_B = (\mathcal{B}+1)^2$ and above values, we have with probability at least $1 - \exp\{-n^{d/(d+4\beta)}\}$,

$$
\begin{aligned}
R(\hat{f}_n) - R(f_0) \leq {} & (\log n)^2 n^{-\beta/(3d/8 + 2\beta)} \\
& \times \left\{c(\beta+1)^8(\mathcal{B}+1)^2 d^{3\lfloor \beta \rfloor + 3}/(s_{\min} - 1) + 18^2 \mathcal{B}^2(\lfloor \beta \rfloor + 1)^4 d^{2\lfloor \beta \rfloor + (\beta \vee 1)}\right\},
\end{aligned}
$$

or simply

$$
R(\hat{f}_n) - R(f_0) \leq C(d, \beta, \mathcal{B}, s_{\min})(\log n)^2 n^{-\beta/(3d/8 + 2\beta)},
$$

where $C(d, \beta, \mathcal{B}, s_{\min}) = O(\mathcal{B}^2(\lfloor \beta \rfloor + 1)^8 d^{3\lfloor \beta \rfloor + 3}/(s_{\min} - 1))$ is a constant independent of $n$. For the induced classifier $\hat{h}_n = \mathrm{sign}(\hat{f}_n)$, the excess misclassification error satisfies

$$
R_*(\hat{h}_n) - R_*(h_0) \leq \sqrt{C(d, \beta, B_0, s_{\min})}(\log n) n^{-4\beta/(3d + 16\beta)}.
$$

The excess risk bound under the approximate low-dimensional manifold assumption can be obtained in a similar way.

## 3.5 Excess misclassification errors

In this subsection, we summarize the misclassification error bounds obtained in existing results and compare them with ours.

When the hypothesis space is taken to be the class of measurable functions itself, for the excess misclassification error with the 0-1 loss, the convergence rates of order $O(n^{-1/2})$ can be attained

using oracle inequalities [20, 26]. Under Tsybakov noise condition of exponent $\theta$ and $\alpha$-Hölder smooth condition on the decision boundaries, [24] proved that the minimax lower bound of the excess misclassification rate is $O(n^{-\alpha(\theta+1)/\{\alpha(\theta+2)+(d-1)\theta\}})$ over all measurable classifiers. With large enough $\alpha$ and $\theta$, the convergence rate can be close to $O(n^{-1})$ arbitrarily.

For classifications using deep neural networks with the SVM hinge loss function, [16] showed that the minimax optimal convergence rate is $O(n^{-\alpha(\theta+1)/\{\alpha(\theta+2)+(d-1)\theta\}})$ under the Tsybakov noise condition of exponent $\theta$ and $\alpha$-Hölder smooth condition on the decision boundaries and $O(n^{-\alpha(\theta+1)/\{\alpha(\theta+2)+d\}})$ under the Tsybakov noise condition of exponent $\theta$ and $\alpha$-Hölder smooth condition on the conditional class probability $\eta(x) = P(Y=1|X=x)$.

For classifications using deep convolutional networks with the $p$-norm hinge loss function $\phi(u) = \max\{1-u,0\}^p$ with $p \geq 1$ (it is hinge loss when $p=1$), [9] derived the approximation error rate and the excess misclassification rate for a target function in the Sobolev space $W^{\beta,p}(\mathbb{S}^{d-1})$ with input data supported on the sphere $\mathbb{S}^{d-1}$ in $\mathbb{R}^d$ under the variancing power condition [25]. Two quantities including $\gamma = \max\{1, (d+3+\beta)/2(d-1)\}$ and $\tau$ for variancing power condition are involved in the convergence rate. Details can be found in [9]. Based on Table II in [9], we list the excess misclassification rates and compare them in the table below.

Table 3: Excess Misclassification Error

| Hypothesis space | Loss | Condition | Rate | Reference |
|---|---|---|---|---|
| Measurable functions | 0-1 loss | $\theta$-noise condition; $\alpha$-Hölder decision boundary | $O(n^{-\frac{\beta(\theta+1)}{\beta(\theta+2)+(d-1)\theta}})$ | Theorem 1 in [24] |
| DNN | Hinge | | $O(n^{-\frac{\beta(\theta+1)}{\beta(\theta+2)+(d-1)(\theta+1)}})$ | Theorem 1 in [16] |
| Deep CNNs | 1-norm | $f_0 \in W^{\beta,p}(\mathbb{S}^{d-1})$ | $O(n^{-\frac{\beta}{\beta(2-\tau)+2\gamma(d-1)}})$ | Theorem 2 in [9] |
| | p-norm | | $O(n^{-\frac{p\beta}{2p\beta(2-\tau)+2p(\gamma+1)(d-1)}})$ | |
| | 2-norm | $f_0 \in W^{\beta,p}(\mathbb{S}^{d-1})$; $\theta$-noise condition; | $O(n^{-\frac{2\beta\theta}{(2+\theta)((\gamma+1)(d-1)+2\beta)}})$ | Theorem 3 in [9] |
| | Hinge | $\theta$-noise condition; $f_0 \in W^{\beta,p}([0,1]^d)$ | $O(n^{-\frac{\beta(\theta+1)}{d+2\beta(\theta+1)}})$ | This paper |
| | Logistic | $f_0 \in W^{\beta,p}([0,1]^d)$ | $O(n^{-\frac{\beta}{2d+4\beta}})$ | |
| | Exponential | $f_0 \in W^{\beta,p}([0,1]^d)$ | $O(n^{-\frac{\beta}{2d+4\beta}})$ | |
| | Least square | $f_0 \in W^{\beta,p}([0,1]^d)$ | $O(n^{-\frac{4\beta}{3d+16\beta}})$ | |

## 4  A toy example for CNN approximation

In this subsection, we use a toy example to illustrate the approximation power of CNNs. We examine how the approximation error varies according to the filter size and depth of the CNNs.

We consider a target function $f_0$ defined as

$$f_0(x) = 2\sin(2\pi x_1) + 4(x_2)^3, \ x \in [0,1]^2$$

where $x = (x_1, x_2) \in [0,1]^2$ is the two-dimensional input. The 2D heatmap and 3D surface visualizations for the target function $f_0$ are presented in Figure 1.

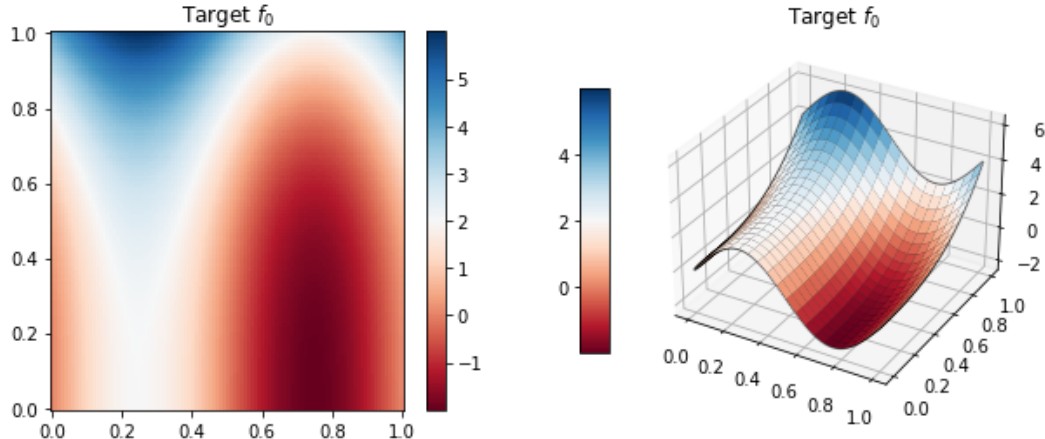

Figure 1: Heatmap and 3D surface plots for the target function $f_0$ defined on $[0, 1]^2$.

To construct CNNs that approximates $f_0$, we train the CNNs on noiseless data, which is generated by taking values of the target function in $[0, 1]^2$. Specifically, we choose $100 \times 100$ lattice points $\{X_i\}_{i=1}^{10,000}$ uniformly on the square $[0, 1]^2$ and calculate the corresponding target function values $\{Y_i = f_0(X_i)\}_{i=1}^{10,000}$ to get the training data $\{X_i, Y_i\}_{i=1}^{10,000}$ for function approximation.

We implement the training in Python via *Pytorch* and use *Adam* [17] as the optimization algorithm with default learning rate 0.01 and default $\beta = (0.9, 0.99)$ (coefficients used for computing running averages of gradients and their squares). During the training process, we set the batch size as $2, 500$, i.e. the $1/4$ of the sample size. The maximum epoch of training is set as $1,000$ and an early stopping rule is applied where we stop the iteration if the training losses for the last 200 consecutive epochs do not achieve a new minimum.

We investigate the approximation power of CNNs with different shapes, which are specified through the filter length and the depth. For ease of demonstration, we consider CNNs with $L$ hidden layers with each hidden layer being a convolutional layer with a uniform filter length $s$. We set the last layer as a fully connected layer with one-dimensional scalar output.

We train 12 CNNs that approximate the target function for $L = 1, 2, 3$ and $s = 20, 50, 100, 200$, then we calculate the empirical $L_1$ and $L_2$ distances measuring the approximation error according to

$$L_1(\hat{f}, f_0) = \frac{1}{10,000} \sum_{i=1}^{10,000} |\hat{f}(X_i) - f_0(X_i)|,$$

$$L_2(\hat{f}, f_0) = \sqrt{\frac{1}{10,000} \sum_{i=1}^{10,000} |\hat{f}(X_i) - f_0(X_i)|^2},$$

where $\hat{f}$ denotes the approximate function by CNN. Due to the randomness of the optimization algorithms, we train each CNN for several times and take the best one as the final approximator on the target function.

The summary statistics of empirical $L_1$ and $L_2$ distances between the CNNs to the target function on the training data are presented in Table 4. We can see that the numerical results generally support our theory in the sense that the approximation error shrinks by a proper rate with respect to the filter length and depth of CNN. However, we would like to mention that the optimization procedure using *Adam* is not fully tuned and optimization errors may slightly perturb the results, especially when the network is deep and wide.

A visualization of CNN approximations on the target function by heatmaps are presented in Figure 2.

Table 4: Approximation errors by CNNs with different filter lengths and depths.

| Approximation error $L_1(L_2)$ | | Filter length | | | |
|---|---|---|---|---|---|
| | | 20 | 50 | 100 | 200 |
| Hidden layers | 1 | 0.807(0.969) | 0.450(0.539) | 0.139(0.186) | 0.062(0.084) |
| | 2 | 0.112(0.144) | 0.055(0.070) | 0.047(0.064) | 0.025(0.037) |
| | 3 | 0.078(0.098) | 0.051(0.070) | 0.037(0.046) | 0.032(0.045) |

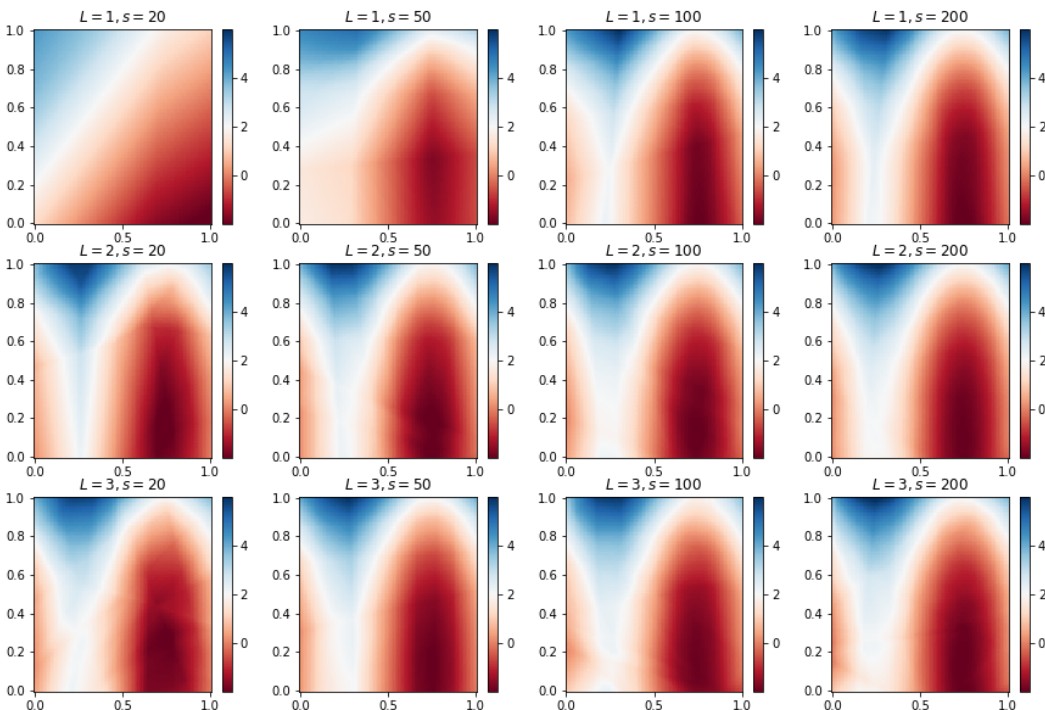

Figure 2: Heatmaps for the CNN approximations on the target function. The CNNs are designated with depth $L = 1, 2, 3$ and filter length $s = 20, 50, 100, 200$.

# 5 Proofs

In this section, we prove Theorems 2.1, 2.2, 2.4, 4.4, 4.6, 4.7, as well as the non-asymptotic error bound results for the cross entropy, the SVM, the logistic, the exponential and the least squares losses.

## 5.1 Proof of Theorem 2.1

As in [13], this approximation result in term of $\| \cdot \|_{W^{\beta-1,p}}$ norm is proved for target function $f \in W^{\beta,p}(\mathcal{X})$, which can be seen as an generalization of Theorem 1 in [27] where $L^\infty$ error bounds for $f \in W^{\beta,\infty}(\mathcal{X})$ is derived. Besides, our results improves the prefactor in $d$ of the network width in Theorem 4.1 in [13]. The main idea of our proof is to approximate the averaged Taylor expansion of function $f$ in Sobolev functions. The definition and properties of averaged Taylor expansion can be found in subsection 5.13 of this supplementary material. By Lemma 5.6, let $B$ be a ball with proper radius in $\Omega$ such that $\Omega$ is star-shaped with respect to $B$, for $1 \le p \le \infty$ we have

$$\|f - T^\beta f\|_{W^{m,p}(\Omega)} \le C_{m,\beta,\gamma} r_\Omega^{\beta-m} \|f\|_{W^{m,p}(\Omega)}, \qquad k = 0, 1, \ldots, \beta$$

where $C_{k,\beta,\gamma} > 0$ is a constant, $r_\Omega$ is the diameter of $\Omega \subseteq \mathcal{X}$, $\gamma$ is the chunkiness parameter of $\Omega$ defined in definition 5.4 and $T^\beta f$ is the averaged Taylor expansion over $B$ defined as in (13):

$$T^\beta f(x) = \int_B T_y^\beta f(x) \phi(y) dy,$$

where

$$T_y^\beta f(x) = \sum_{\|\alpha\|_1 \leq \beta} \frac{1}{\alpha!} D^\alpha f(y)(x-y)^\alpha,$$

and $\phi$ is an arbitrary cut-off function supported in $\bar{B}$ being infinitely differentiable.

This reminder term could be well controlled when the approximation to Taylor expansion in implemented in a fairly small local region. Then we can focus on the approximation of the Taylor expansion locally. The proof is divided into four parts:

(i) Partition $\mathcal{X}$ into small cubes $\cup_\theta Q_\theta$, and construct a network $\psi$ that approximately maps each $x \in Q_\theta$ to a fixed point $x_\theta \in Q_\theta$. Hence, $\psi$ approximately discretize $\mathcal{X}$.

(ii) For any multi-index $\alpha$, construct a network $\phi_\alpha$ that approximates the coefficient of the averaged Taylor expansion over a ball $B$ in $Q_\theta$, that is $x \in Q_\theta \mapsto c_\alpha(x_\theta)$, where $Q_\theta = \sum_{\alpha \leq \beta} c_\alpha(x_\theta) x^\alpha$ by Lemma 5.2. Once $\mathcal{X}$ is discretized, the approximation is reduced to a data fitting problem.

(iii) Construct a network $P_\alpha(x)$ to approximate the polynomial $x^\alpha := x_1^{\alpha_1} \dots x_d^{\alpha_d}$ where $x = (x_1, \dots, x_d)^\top \in \mathbb{R}^d$ and $\alpha = (\alpha_1, \dots, \alpha_d)^\top \in \mathbb{N}_0^d$. In particular, we can construct a network $\phi_\times(\cdot, \cdot)$ approximating the product function of two scalar inputs.

(iv) Lastly, our construction of neural network can be written in the form,

$$\phi(x) = \sum_{\|\alpha\|_1 \leq \beta} \phi_\times \Big( \phi_\alpha(x), P_\alpha(x) \Big).$$

*Proof.* Before proving the theorem, we firstly show that each ReLU activated multilayer-perceptron can be computed by a downsampled CNN $f_{CNN}$ defined as in (4). The proof here follows Theorem 2 in [29] where we construct downsampled CNN exactly computing a fully-connected layer. Due to the simplicity of the formulation of ReLU activated multilayer-perceptron, we focus on showing that each fully-connected layer can be reproduced by a down sampled CNN with proper choice of filters. Without loss of generality, let

$$W = \begin{bmatrix} w_{11} & w_{12} & \cdots & w_{1d_{in}} \\ w_{21} & w_{22} & \cdots & w_{2d_{in}} \\ \vdots & \cdots & \cdots & \vdots \\ w_{d_{out}1} & w_{d_{out}2} & \cdots & w_{d_{out}d_{in}} \end{bmatrix},$$

be the weight matrix and $b = (b_1, \dots, b_{d_{out}})^\top \in \mathbb{R}^{d_{out}}$ be the bias vector of a fully-connect layer $A_F(\cdot) : \mathbb{R}^{d_{in}} \to \mathbb{R}^{d_{out}}$, where $w_{ij}, i = 1, \dots, d_{out}; j = 1, \dots, d_{in}$ are weights. Then the output of the fully-connect layer with respect to $x \in \mathbb{R}^{d_{in}}$ is

$$A_F(x) = \sigma(Wx + b).$$

To construct a downsampled CNN to represent the fully-connected layer, we define a sequence $w = (w_0, \dots, w_{d_{in} \times d_{out}}) \in \mathbb{R}^{d_{in} \times d_{out}}$ with length $d_{in} \times d_{out}$ by stacking the reversed row vectors of the weight matrix $W$ as

$$w_{i+(r-1)d_{in}} = W_{r,d_{in}-i}, \quad r = 1, 2, \dots, d_{out}, \ i = 0, 1, \dots, d_{in} - 1,$$

or

$$w = (w_{1d_{in}}, \dots, w_{11}, w_{2d_{in}}, \dots, w_{21}, \dots, w_{d_{out}d_{in}}, \dots, w_{d_{out}1}),$$

where $w_{ij}$ are weights of the matrix $W$. By Lemma 2 in [29], there exists a sequence of filter $\{w^{(i)}\}_{i=1}^I$ of equal filter length $s$ for some $s \in [2, d_{in} \times d_{out}]$ and $I \leq \lceil (d_{in} \times d_{out} - 1)/(s-1) \rceil$ such that $w$ has convolutional factorization $w^{(I)} * w^{(I-1)} * \cdots * w^{(1)}$. We let $\{w^{(i)}\}_{i=1}^I$ be the filters of the $I$ Toeplitz type convolutional matrices $T^{(1)}, \dots, T^{(I)}$ of the convolutional operators

$$A^{(i)}(x) = \sigma(T^{(i)}x + b^{(i)})$$

for $i = 1, \ldots, I$. Next we construct the bias vectors $\{b^{(i)}\}_{i=1}^{I}$ according to Lemma 3 in [29]. Let

$$b^{(1)} = -B\|w^{(1)}\|_1 \mathbb{1}_{d_{in}+s}$$

with $B$ denoting the infinity norm of the input $x$ of the fully-connected layer, and let

$$b^{(i)} = -B(\Pi_{j=1}^{i-1}\|w^{(j)}\|_1)T^{(j)}\mathbb{1}_{d_{in}+s(i-1)} + B(\Pi_{j=1}^{i}\|w^{(j)}\|_1)\mathbb{1}_{d_{in}+is}$$

for $i = 2, \ldots, I-1$. Then we have

$$A^{(I-1)} \circ \cdots \circ A^{(1)}(x) = T^{(i-1)}\cdots T^{(1)}x + B(\Pi_{j=1}^{I-1}\|w^{(j)}\|_1)\mathbb{1}_{d_{in}}.$$

Finally, we set

$$b^{(i)} = \begin{cases} -B(\Pi_{j=1}^{I-1}\|w^{(j)}\|_1)T^{(j)}\mathbb{1}_{d_{in}+s(I-1)} + \theta, & \text{if } I \geq 2, \\ \theta, & \text{if } I = 1, \end{cases}$$

where $\theta \in \mathbb{R}^{d_{in}+s(I-1)}$ s an arbitrary vector satisfying $D_I(\theta) = b$ with $D_I : \mathbb{R}^{d_{in}+sI} \to \mathbb{R}^{d_{out}}$ denoting the downsampling operator and $b$ is the bias vector of the target fully-connected layer. Then we can defined the downsampled CNN by

$$f_{CNN} = D_I \circ A^{(I)} \circ \cdots \circ A^{(1)},$$

and by Lemma 1 in [29] we have

$$f_{CNN}(x) = D_I(\sigma(T^{(I)}\cdots T^{(1)}x + \theta)) = \sigma(Wx + b).$$

And the the total number of free parameters in computing the fully-connected layer $A_F$ is no more than

$$(3s+2)\lceil\frac{(d_{in} \times d_{out} - 1)}{(s-1)}\rceil \leq 8d_{in} \times d_{out},$$

which is 8 times of the total parameters of the fully-connected layer. The depth of $f_{CNN}$ is $I + 1 \leq \lceil(d_{in} \times d_{out} - 1)/(s-1)\rceil + 1$. If we take $s = d_{in} \times d_{out}$, then $f_{CNN}$ is a 2-layer convolutional network. By induction, we know that for any MLP with depth $L$, width $(d_0, d_1, \ldots, d_{L+1})$ and size $\mathcal{S}$, there exists a downsampled CNN as defined in (4) that exactly computes the MLP. The downsampled CNN has size no more than $8\mathcal{S}$, depth no more than $L + \sum_{i=0}^{L}\lceil d_i \times d_{i+1}/(s^{(i)}-1)\rceil$ where $s^{(i)}$ is the filter length of the $2i$-th layer of downsampled CNN and the $2i-1$-th layers are downsampling operations. More specifically, if take the maximum possible filter size $s^{(i)} = d_i \times d_{i+1}$ for each construction of the fully-connected layer above, the downsampled CNN that computed the MLP has depth no more than $2L$, size no more than $8\mathcal{S}$ and maximum filter length no more than $\mathcal{W} = \max\{d_0 d_1, \ldots, d_L d_{L+1}\}$. In general, the maximum filter length $s = \mathcal{O}(1)$, and the constructed downsampled CNN has size no more than $8\mathcal{S}$, depth no more than $L + \sum_{i=0}^{L}\lceil d_i \times d_{i+1}/(s^{(i)}-1)\rceil \leq L(1 + \mathcal{W}^2/(s_{\min}-1)) \leq 2L\mathcal{W}^2/(s_{\min}-1)$ where $s_{\min}$ is the minimum of the filter lengths over convolutional layers.

Now we prove the theorem following the four parts described above. Part (i) (domain discretization) uses an idea similar to that in the proof of Theorem 3.3 in [15]. We target for approximation on Sobolev functions that the inequalities will be obtained for Sobolev norms instead of $L_1$ or $L_2$ norms, and the local averaged Taylor's expansion instead of Taylor's expansion of the target function will be approximated.

Without loss of generality, we assume the Sobolev norm of $f$ is 1, i.e. $\|f\|_{W^{\beta,p}(\mathcal{X})} \leq 1$. The reason is that we can always approximate $f/B_0$ firstly by a network $\phi$ with approximation error $\epsilon$, then the scaled network $B_0\phi$ will approximate $f$ with error no more than $\epsilon B_0$. Besides, it is a trivial case when the Sobolev norm of $f$ is 0. Next, we prove the four parts of the sketch as follows.

**Part (i):** Discretization.
We first divide the domain $\mathcal{X}$ approximately into hypercubes as in [23, 15]. We present below the sketch, and details can be found in proof of Theorem 3.3 in [15].

(i) The domain is approximately divided into hypercubes $\{Q_\theta\}$ indexed by $\theta$. Given $N, M \in \mathbb{N}^+$, set $K = \lfloor N^{1/d}\rfloor^2\lfloor M^{2/d}\rfloor$ and $\delta \in (0, 1/(3K)]$, for each $\theta = (\theta_1, \ldots, \theta_d) \in \{0, 1, \ldots, K-1\}^d$, we define

$$Q_\theta := \Big\{x = (x_1, \ldots, x_d) : x_i \in [\frac{\theta_i}{K}, \frac{\theta_i + 1}{K} - \delta \cdot 1_{\theta_i < K-1}], i = 1, \ldots, d\Big\}.$$

Let $\Omega(\mathcal{X}, K, \delta) = \mathcal{X}\backslash \cup_\theta Q_\theta$ be the trifling region or the complement of $\cup_\theta Q_\theta$.

(ii) For each $\theta$, there exists a ReLU network maps all $x \in Q_\theta$ to a common value depending on $\theta$. To be specific, by Lemma 5.7 and proof in Theorem 3.3 in [15], there exists a ReLU network $\psi(x)$ with width $d(4\lfloor N^{1/d} \rfloor + 3)$ and depth $4M + 5$ such that for any $\theta = (\theta_1, \ldots, \theta_d) \in \{0, 1, \ldots, K-1\}^d$, we have

$$\psi(x) = \psi(x_1, \ldots, x_d) = \theta/K := (\theta_1/K, \ldots, \theta_d/K)^\top$$

for all $x = (x_1, \ldots, x_d)^\top \in Q_\theta$.

**Part (ii):** Local Approximation of averaged Taylor expansion coefficients.
We consider the local approximation of $f \in W^{\beta, p}(\mathcal{X})$ on the subset $Q_\theta$ for each $\theta$ by its averaged Taylor expansion $T^\beta f(x) = \int_{B_\theta} T_y^\beta f(x)\phi(y)dy$ over the ball $B_\theta$ where $T_y^\beta f(x) = \sum_{\|\alpha\|_1 \leq \beta} \frac{1}{\alpha!} D^\alpha f(y)(x-y)^\alpha$, the ball $B_\theta = \{y \in \mathcal{X} : \|y - (\theta + (1-\delta)/2)/K\|_2 \leq (1-\delta)/(2K)\}$ support in $Q_\theta$, and $\phi$ is an cut-off function supported in $\bar{B}$ defined in Definition 5.1. Note that $Q_\theta$ has diameter $r_{Q_\theta} = \sqrt{d}(1-\delta)/K$ and it is star-shaped with respect to the ball $B_\theta$ with radius $(1-\delta)/(2K)$. By Lemma 5.6, we have error bound for the remainder of the averaged Taylor expansion,

$$\|f - T^\beta f\|_{W^{m,p}(Q_\theta)} \leq C_{m,\beta,\gamma} \left( \frac{\sqrt{d}(1-\delta)}{K} \right)^{\beta - m} \|f\|_{W^{\beta,p}(Q_\theta)} \qquad m = 0, 1, \ldots, \beta.$$

Besides, by Lemma 5.2, the averaged Taylor expansion is actually a polynomial $T^\beta f(x) = \sum_{\|\alpha\|_1 \leq \beta} c_\alpha^\theta x^\alpha$ for $x \in Q_\theta$ with $|c_\alpha^\theta| \leq C(\beta, d)$ for all $\alpha$ where

$$C(\beta, d) = (\beta + 1)2^{\beta + d/p} d^{3\beta/2} (\pi^{-d/2}\Gamma(d/2 + 1))^{1/p} (NM)^{2/p - 2\beta/d} \|f\|_{W^{\beta,p}(\Omega)}.$$

Next we seek to approximate the polynomial $T^\beta f(x) = \sum_{\|\alpha\|_1 \leq \beta} c_\alpha^\theta x^\alpha$ locally by deep neural networks. In this part of proof, we firstly construct network to approximate the coefficients of averaged Taylor expansion.

For each hypercube $Q_\theta$, we locate the input $x \in Q_\theta$ by mapping multi-dimensional $x$ to an integer $i_\theta$ for each $\theta$. Given the fact that $\theta \in \{0, 1, \ldots, K-1\}^d$ is one-to-one correspondence to $i_\theta := \sum_{j=1}^d \theta_j K^{j-1} \in \{0, 1 \ldots, K^d - 1\}$, and based on the locator $\psi$ in part (i), it is proved that in [19, 15], there exists a network $\psi_0 : \mathbb{R}^d \to \mathbb{R}$ with width $d(4\lfloor N^{1/d} \rfloor + 3)$ and depth $4M + 5$ such that for each $\theta \in \{0, 1, \ldots, K-1\}^d$, $\psi_0(x) = \sum_{j=1}^d \theta_j K^{j-1} = i_\theta$ for all $x \in Q_\theta$.

For any $\alpha \in \mathbb{N}_0^d$ satisfing $\|\alpha\|_1 \leq \beta$ and each $i = i_\theta \in \{0, 1, \ldots, K^d - 1\}$, we denote $\xi_{\alpha, i} := (c_\alpha^\theta + C(\beta, d))/(2C(\beta, d)) \in [0, 1]$. Since $K^d \leq N^2 M^2$, by Lemma 5.8, there exists a ReLU network $\varphi_\alpha$ with width $16(\beta + 2)(N + 1)\lceil \log_2(8N) \rceil$ and depth $5(M + 2)\lceil \log_2(4M) \rceil$ such that

$$|\varphi_\alpha(i) - \xi_{\alpha, i}| \leq (NM)^{-2\beta - 4},$$

for all $i_\theta \in \{0, 1, \ldots, K^d - 1\}$. We define

$$\phi_\alpha(x) := 2C(\beta, d)\varphi_\alpha(\psi_0(x)) - C(\beta, d) \in [-C(\beta, d), C(\beta, d)], \quad x \in \mathbb{R}^d.$$

Then $\phi_\alpha$ can be implemented by a network with width $16d(\beta + 2)(N + 1)\lceil \log_2(8N) \rceil \leq 32d(\beta + 2)N\lceil \log_2(8N) \rceil$ and depth $5(M + 2)\lceil \log_2(4M) \rceil + 4M + 5 \leq 15M\lceil \log_2(8M) \rceil$ such that

$$|\phi_\alpha(x) - c_\alpha^\theta| = 2C(\beta, d)|\varphi_\alpha(i_\theta) - \xi_{\alpha, i_\theta}| \leq 2C(\beta, d)(NM)^{-2\beta - 4}, \tag{5}$$

for any $\theta \in \{0, 1, \ldots, K-1\}^d$ and $x \in Q_\theta$.

**Part (iii):** Local Approximation of the polynomial.
In this part, we present two lemmas showing that the multiplication operator and polynomials can be approximated by proper networks under $\|\cdot\|_{W^{1,\infty}}$ norm. For multiplication operator, by Lemma 5.9, there exists a ReLU network with width $9N + 1$ and depth $4\beta M + 8$ such that for any $t_1, t_2 \in [a, b]$,

$$\|t_1 t_2 - \phi_\times(t_1, t_2)\|_{W^{1,\infty}([a,b]^2)} \leq 6(b-a)^2 N^{-2\beta M - 4}, \tag{6}$$

with $\|\phi_\times\|_{W^{1,\infty}((a,b)^2)} \leq 12(b-a)^2$. For polynomials, by Lemma 5.10, for any $\alpha \in \mathbb{N}_0^d$ with $\|\alpha\|_1 \leq \beta \in \mathbb{N}^+$, there exists a ReLU network $P_\alpha$ with width $9(N + 1) + \beta$ and depth $7(\beta + 1)^2 M$ and $\|P_\alpha\|_{W^{1,\infty}(\mathcal{X})} \leq 18$ such that for any $x \in \subseteq [0, 1]^d$

$$\|P_\alpha(x) - x^\alpha\|_{W^{1,\infty}(\mathcal{X})} \leq 10(\beta + 1)(N + 1)^{-7(\beta + 1)M}. \tag{7}$$

Note that above approximation result holds for $x \in [0,1]^d$, we then provide below network construction for truncation operator of the input onto $[0,1]^d$. Let $\varphi_0(t) = \min\{\max\{t,0\},1\} = \sigma(t) - \sigma(t-1)$ for $t \in \mathbb{R}$, where $\sigma(\cdot)$ is the ReLU activation function. And let $\varphi(x) = \varphi(x_1, \ldots, x_d) := (\varphi_0(x_1), \ldots, \varphi_0(x_d))^\top$ denote the extension of $\varphi_0$ to $\mathbb{R}^d$. Then $\varphi$ truncates the input $x \in \mathbb{R}^d$ onto $[0,1]^d$.

**Part (iv):** Approximation of $f$ on $\cup_{\theta \in \{0,1,\ldots,K-1\}^d} Q_\theta$.

For any $x \in Q_\theta$, $\theta \in \{0,1,\ldots,K-1\}^d$, we can now approximate the averaged Taylor expansion of $f(x)$ by combined sub-networks. Motivated by this, we define

$$\phi_0(x) := \sum_{\|\alpha\|_1 \leq \beta} \phi_\times \Big(\phi_\alpha(x), P_\alpha(\varphi(x))\Big) \in [-d^\beta C(\beta,d), d^\beta C(\beta,d)],$$

where $\|\phi_0\|_{W^{1,\infty}(\mathcal{X})} \leq 48 C(\beta,d)^2 (\beta+1) d^\beta$. Then $\phi_0$ is our constructed neural network approximating the target function $f$ on $\mathcal{X}$. Recall that width and depth of $\varphi$ is $(2d, 1)$, width and depth of $\psi$ is $(d(4\lfloor N^{1/d}\rfloor + 3), 4M + 5)$, width and depth of $P_\alpha$ is $(9(N+1) + \beta, 7(\beta+1)^2 M)$, width and depth of $\phi_\alpha$ is width $(16d(\beta+2)(N+1)\lceil\log_2(8N)\rceil, 5(M+2)\lceil\log_2(4M)\rceil + 4M + 5)$ and width and depth of $\phi_\times$ is $(9N+1, 2\beta M + 4)$. Hence, by our construction, $\phi_0$ can be implemented by a neural network with width $38(\beta+1)^2 d^{\beta+1} N\lceil\log_2(8N)\rceil$ and depth $21(\beta+1)^2 M\lceil\log_2(8M)\rceil$.

For $\theta \in \{0,1,\ldots,K-1\}^d$, thanks to Lemma 5.6, we have error bound for the remainder of the averaged Taylor expansion,

$$\|f - T^\beta f\|_{W^{m,p}(Q_\theta)} \leq C_{\beta,d,r}\Big(\frac{\sqrt{d}(1-\delta)}{K}\Big)^{\beta-m}\|f\|_{W^{\beta,p}(Q_\theta)},$$

where $C_{\beta,d,r} = 3\beta d^{\beta-1}\Gamma(d/2+1)\pi^{-d/2}(1+2\sqrt{d})^d$. Then for any $\theta \in \{0,1,\ldots,K-1\}^d$, the approximation error of $\phi_0$ on $f$ over $Q^\theta$ can be upper bounded,

$$\|\phi_0 - f\|_{W^{1,p}(Q_\theta)} \leq \|\phi_0 - T^\beta f\|_{W^{1,p}(Q_\theta)} + \|T^\beta f - f\|_{W^{1,p}(Q_\theta)}$$

$$\leq \|\sum_{\|\alpha\|_1 \leq \beta} \phi_\times\Big(\phi_\alpha(x), P_\alpha(\varphi(x))\Big) - c_\alpha^\theta x^\alpha\|_{W^{1,\infty}(Q^\theta)}$$

$$+ C_{m,\beta,\gamma}\Big(\frac{\sqrt{d}(1-\delta)}{K}\Big)^{\beta-m}\|f\|_{W^{\beta,p}(Q_\theta)}. \tag{8}$$

We denote $\mathcal{E}_\alpha = \|c_\alpha^\theta x^\alpha - \phi_\times\big(\phi_\alpha(x), P_\alpha(\varphi(x))\big)\|_{W^{1,\infty}(Q_\theta)}$ for each $\alpha \in \mathbb{N}_0^d$ with $\|\alpha\|_1 \leq \beta$. Using the inequality $|t_1 t_2 - \phi_\times(t_3, t_4)| \leq |t_1 t_2 - t_3 t_2| + |t_3 t_2 - t_3 t_4| + |t_3 t_4 - \phi_\times(t_3, t_4)| \leq |t_2||t_1 - t_3| + |t_3||t_2 - t_4| + |t_3 t_4 - \phi_\times(t_3, t_4)|$ for any $t_1, t_2, t_3, t_4 \in \mathbb{R}$, and by (5), (6) and (7), for $\|\alpha\|_1 \leq \beta$ we have

$$\mathcal{E}_\alpha \leq |x^\alpha|\|c_\alpha^\theta - \phi_\alpha(x)\|_{W^{1,\infty}(Q_\theta)} + |\phi_\alpha(x)|\|x^\alpha - P_\alpha(\varphi(x))\|_{W^{1,\infty}(Q_\theta)}$$

$$+ \Big\|\phi_\alpha(x)P_\alpha(\varphi(x)) - \phi_\times\big(\phi_\alpha(x), P_\alpha(\varphi(x))\big)\Big\|_{W^{1,\infty}(Q_\theta)}$$

$$\leq 2C(\beta,d)(NM)^{-2\beta-4} + 10C(\beta,d)(\beta+1)(N+1)^{-7\beta M-7M} + 24C(\beta,d)^2 N^{-2\beta M-4}$$

$$\leq (12C(\beta,d)(\beta+1) + 24C(\beta,d)^2)(NM)^{-2\beta-4}$$

$$\leq 36(\beta+1)^2 2^{2\beta+2d/p} d^{3\beta}(\pi^{-d/2}\Gamma(d/2+1))^{2/p} B_0\|f\|_{W^{\beta,p}(\Omega)}(NM)^{-2\beta-4\beta/d}, \tag{9}$$

where the last inequality holds since $p \geq 1$ and $\|f\|_{W^{\beta,p}(\Omega)} \leq B_0$. Moreover, as in the proof of Theorem 3.3 in [15], the number of terms $|\{\alpha : \alpha \in \mathbb{N}_0^d, \|\alpha\|_1 \leq \beta\}|$ in the summation can be bounded by $(\beta+1)d^\beta$.

Recall that $K = \lfloor N^{1/d}\rfloor^2 \lfloor M^{2/d}\rfloor$, combining (8) and (9), we have

$$\|\phi_0 - f\|_{W^{1,p}(Q_\theta)} \leq \|\phi_0 - T^\beta f\|_{W^{1,p}(Q_\theta)} + \|T^\beta f - f\|_{W^{1,p}(Q_\theta)}$$

$$\leq 36 \cdot 2^{2\beta+2d/p}(\beta+1)^3 d^{4\beta}(\pi^{-d/2}\Gamma(d/2+1))^{2/p} B_0\|f\|_{W^{\beta,p}(\Omega)}(NM)^{-2\beta-4\beta/d}$$

$$+ \beta d^{3\beta/2}\Gamma(d/2+1)\pi^{-d/2}(1+1/r)^d\|f\|_{W^{\beta,p}(Q_\theta)}(NM)^{-2(\beta-m)/d}$$

$$\leq 37 \cdot 2^{2\beta+2d/p}(1+2\sqrt{d})^d(\beta+1)^3 d^{4\beta}(\pi^{-d/2}\Gamma(d/2+1))^{2/p+1}$$

$$\times B_0\|f\|_{W^{\beta,p}(Q_\theta)}(NM)^{-2(\beta-m)/d}, \tag{10}$$

for $\theta \in \{0, 1, \ldots, K-1\}^d$. Note that $\mathcal{X} = \Omega(\mathcal{X}, K, \delta) \cup (\cup_{\theta \in \{0,1,\ldots,K-1\}^d} Q_\theta)$ and the Lebesgue measure of $\Omega(\mathcal{X}, K, \delta)$ can be upper bounded by $\delta dK$ since $\delta \in (0, 1/(3K)]$ can be arbitrarily small. Then the approximation error $\|f(x) - \phi_0(x)\|_{W^{1,p}(\mathcal{X})}$ can be further bounded as follows.

$$\|f(x) - \phi_0(x)\|_{W^{1,p}(\mathcal{X})} = \Big( \sum_{\theta \in \{0,1,\ldots,K-1\}^d} \|f - \phi_0\|_{W^{1,p}(Q_\theta)}^p + \|f - \phi_0\|_{W^{1,p}(\Omega(\mathcal{X},K,\delta))}^p \Big)^{1/p}$$

$$\leq 37 \cdot B_0 2^{2\beta+2d/p}(1+2\sqrt{d})^d (\beta+1)^3 d^{4\beta} (\pi^{-d/2}\Gamma(d/2+1))^{2/p+1}(NM)^{-2(\beta-m)/d}$$

$$\times \Big( \sum_{\theta \in \{0,1,\ldots,K-1\}^d} \|f\|_{W^{\beta,p}(Q_\theta)}^p \Big)^{1/p}$$

$$\leq 37 \cdot 2^{2\beta+2d/p}(1+2\sqrt{d})^d (\beta+1)^3 d^{4\beta} (\pi^{-d/2}\Gamma(d/2+1))^{2/p+1}(NM)^{-2(\beta-m)/d}$$

$$\times B_0 \|f\|_{W^{\beta,p}(\mathcal{X})}$$

$$= 37 \cdot 2^{2\beta+2d/p} B_0^2 (\beta+1)^3 (\pi^{-d/2}\Gamma(d/2+1))^{2/p+1}(1+2\sqrt{d})^d d^{4\beta}(NM)^{-2(\beta-m)/d},$$

where $\|f - \phi_0\|_{W^{1,p}(\Omega(\mathcal{X},K,\delta))}^p$ can be arbitrarily small since $\|f\|_{W^{1,\infty}(\mathcal{X})}^p$ and $\|\phi_0\|_{W^{1,\infty}(\mathcal{X})}^p$ are bounded, and $\delta \in (0, 1(3K)]$ can be arbitrarily small.

When $p = \infty$, the target function $f \in W^{\beta,\infty}(\mathcal{X})$ actually is Hölder smooth with order $\beta$ and constant $B_0 = \|f\|_{W^{\beta,\infty}(\mathcal{X})}$. In this case, it is shown in Theorem 3.3 of [15] that there exists a function $\phi_0$ implemented by a neural network with width $38(\lfloor\beta\rfloor+1)^2 d^{\lfloor\beta\rfloor+1} N\lceil\log_2(8N)\rceil$ and depth $21(\lfloor\beta\rfloor+1)^2 M\lceil\log_2(8M)\rceil$ such that

$$|f(x) - \phi_0(x)| \leq 18B_0(\lfloor\beta\rfloor+1)^2 d^{\beta+(\beta\vee1)/2}(NM)^{-2\beta/d},$$

for any $x \in \cup_{\theta \in \{0,1,\ldots,K-1\}^d} Q_\theta$. If the probability measure of $X$ is absolutely continuous with respect to the Lebesgue measure, we also have

$$\mathbb{E}|f(X) - \phi_0(X)| \leq 18B_0(\lfloor\beta\rfloor+1)^2 d^{\beta+(\beta\vee1)/2}(NM)^{-2\beta/d}.$$

As we have shown at the beginning of the proof, the multilayer-perceptron $\phi_0$ here can be computed by a downsampled CNN with filter lengths $2 \leq s_{\min} \leq s_{\max} \leq \mathcal{W} = 38^2(\lfloor\beta\rfloor+1)^4 d^{2\lfloor\beta\rfloor+2} N^2 \lceil\log_2(8N)\rceil^2$ and depth $L \leq 42(\lfloor\beta\rfloor+1)^2 M\lceil\log_2(8M)\rceil \mathcal{W}/(s_{\min}-1)$ and size $\mathcal{S} \leq 8\mathcal{W}L$. We know that there exists a $f_{CNN} \in \mathcal{F}_{\text{CNN}}$ such that the approximation results hold. This completes the proof.

$\square$

## 5.2 Proof of Theorem 2.2

*Proof.* Let $K \in \mathbb{N}^+$ and $\tau \in (0, 1/K)$, define a region $\Omega(\mathcal{X}, K, \tau)$ of $\mathcal{X}$ as

$$\Omega(\mathcal{X}, K, \varepsilon) = \cup_{i=1}^d \{x = [x_1, x_2, \ldots, x_d]^T : x_i \in \cup_{k=1}^{K-1}(k/K - \tau, k/K)\}.$$

By Theorem 3.3 of [15], for any $M, N \in \mathbb{N}^+$, there exists a function $f_* \in \mathcal{F}_{MLP}$ with $\|f_*\|_\infty \leq \mathcal{B}$, depth $21(\lfloor\beta\rfloor+1)^2 M\lceil\log_2(8M)\rceil$ and width $38(\lfloor\beta\rfloor+1)^2 d^{\lfloor\beta\rfloor+1} N\lceil\log_2(8N)\rceil$ such that

$$|f_*(x) - f_{0,T}(x)| \leq 18B_0(\lfloor\beta\rfloor+1)^2 d^{\lfloor\beta\rfloor+(\beta\vee1)/2}(NM)^{-2\beta/d}$$

for any $x \in \mathcal{X}\backslash\Omega(\mathcal{X}, K, \tau)$ where $K = \lfloor N^{1/d}\rfloor^2 \lfloor M^{1/d}\rfloor^2$ and $\tau$ is an arbitrary number in $(0, \frac{1}{3K}]$ and $\lfloor a\rfloor$ denotes the largest integer smaller than $a$. Note that the Lebesgue measure of $\Omega(\mathcal{X}, K, \tau)$ is no more than $dK\tau$ which can be arbitrarily small if $\tau$ is arbitrarily small. Since $P$ is absolutely continuous with respect to Lebesgue measure, then we have

$$\mathbb{E}\|f_*(X) - f_{0,T}(X)\|_2 \leq 18B_0(\lfloor\beta\rfloor+1)^2 d^{\lfloor\beta\rfloor+(\beta\vee1)/2}(NM)^{-2\beta/d}.$$

By the proof of Theorem 2.1, any MLP can be computed exactly by a downsampled CNN. Thus there exists a function $f$ implemented by CNN with its depth no more than $L = 42(\lfloor\beta\rfloor+1)^2 M\lceil\log_2(8M)\rceil \mathcal{W}/(s_{\min}-1)$ where $\mathcal{W} = 38^2(\lfloor\beta\rfloor+1)^4 d^{2\lfloor\beta\rfloor+2} N^2 \lceil\log_2(8N)\rceil^2$ and its filter lengths $2 \leq s_{\min} \leq s_{\max} \leq \mathcal{W} = 38^2(\lfloor\beta\rfloor+1)^4 d^{2\lfloor\beta\rfloor+2} N^2 \lceil\log_2(8N)\rceil^2$ such that

$$\mathbb{E}\|f(X) - f_{0,T}(X)\|_2 \leq 18B_0(\lfloor\beta\rfloor+1)^2 d^{\lfloor\beta\rfloor+(\beta\vee1)/2}(NM)^{-2\beta/d}.$$

$\square$

## 5.3 Proof of Theorem 2.4

*Proof.* The idea of the proof is to project the data into a low-dimensional space using a shallow CNN and then approximate the low-dimensional function using another CNN. The idea is similar to that of Theorem 6.1 in [15] and Theorem 1.2 in [23], and the existence of such a projection follows from Theorem 3.1 in [2]. We present a proof sketch of the projection, and details can be completed following the proof of Theorem 6.1 in [15].

(i) There exists a linear projector $A \in \mathbb{R}^{d_\delta \times d}$ that maps a low-dimensional manifold (embedded in a high-dimensional space) to a low-dimensional space almost preserving the distance, i.e. there exists a matrix $A \in \mathbb{R}^{d_\delta \times d}$ such that $AA^T = (d/d_\delta)I_{d_\delta}$ where $I_{d_\delta}$ is an identity matrix of size $d_\delta \times d_\delta$, and

$$(1-\delta)\|x_1 - x_2\|_2 \le \|Ax_1 - Ax_2\|_2 \le (1+\delta)\|x_1 - x_2\|_2,$$

for any $x_1, x_2 \in \mathcal{M}$.

(ii) For the linear map $A$ and $\rho$-neighborhood $\mathcal{M}_\rho$, we have $A(\mathcal{M}_\rho) \subseteq A(\mathcal{X}) \subseteq E :=$ $[-\sqrt{\frac{d}{d_\delta}}, \sqrt{\frac{d}{d_\delta}}]^{d_\delta}$.

(iii) For any $z \in A(\mathcal{M})$, there exists a unique $x \in \mathcal{M}$ such that $Ax = z$. Then for any $z \in A(\mathcal{M})$, define $x_z = \mathcal{SL}(\{x \in \mathcal{M} : Ax = z\})$ where $\mathcal{SL}(\cdot)$ is a set function which returns a unique element of a set. Then $\mathcal{SL} : A(\mathcal{M}) \to \mathcal{M}$ is a differentiable function with the norm of its derivative locates in $[1/(1+\delta), 1/(1-\delta)]$, i.e. $\frac{1}{1+\delta}\|z_1 - z_2\|_2 \le \|x_{z_1} - x_{z_2}\|_2 \le \frac{1}{1-\delta}\|z_1 - z_2\|_2$, for any $z_1, z_2 \in A(\mathcal{M})$.

(iv) For the function $f_0 : [0,1]^d \to \mathbb{R}^1$, define its low-dimensional representation $\tilde{f}_0 : \mathbb{R}^{d_\delta} \to \mathbb{R}^1$ by $\tilde{f}_0(z) = f_0(x_z)$, for any $z \in A(\mathcal{M}) \subseteq \mathbb{R}^{d_\delta}$. Then $\tilde{f}_0 \in \mathcal{H}^\beta(A(\mathcal{M}), B_0/(1-\delta)^\beta)$ since $f_0 \in \mathcal{H}^\beta(\mathcal{X}, B_0)$.

(v) By the extended version of Whitney' extension theorem in [8], there exists a function $\tilde{F}_0 \in \mathcal{H}^\beta(E, B_0/(1-\delta)^\beta)$ such that $\tilde{F}_0(z) = \tilde{f}_0(z)$ for any $z \in A(\mathcal{M})$, given that $\mathcal{M}$ is compact and $A$ is a linear mapping.

Now on $E = [-\sqrt{d/d_\delta}, \sqrt{d/d_\delta}]^{d_\delta}$, we consider approximate $\tilde{F}_0$ by CNN. By Theorem 2.1, for any $N, M \in \mathbb{N}^+$, there exists a function $\tilde{f}_n : \mathbb{R}^{d_\delta} \to \mathbb{R}^1$ implemented by a downsampled CNN with filter lengths $2 \le s_{\min} \le s_{\max} \le \mathcal{W} = 38^2(\lfloor\beta\rfloor + 1)^4 d_\delta^{2\lfloor\beta\rfloor+2} N^2 \lceil\log_2(8N)\rceil^2$ and depth $L = 42(\lfloor\beta\rfloor + 1)^2 M\lceil\log_2(8M)\rceil\mathcal{W}/(s_{\min} - 1)$ such that

$$|\tilde{f}_n(z) - \tilde{F}_0(z)| \le 18\frac{B_0}{(1-\delta)^\beta}(\lfloor\beta\rfloor + 1)^2 d^{1/2} d_\delta^{(3\beta+1)/2}(NM)^{-2\beta/d_\delta},$$

for all $z \in E\backslash\Omega(E)$ where $\Omega(E)$ is a subset of $E$ with an arbitrarily small Lebesgue measure as well as $\Omega := \{x \in \mathcal{M}_\rho : Ax \in \Omega(E)\}$ does. If we define $f_n^* = \tilde{f}_n \circ A$, i.e. $f_n^*(x) = \tilde{f}_n(Ax)$ for $x \in \mathcal{X}$, then $f_n^*$ is also a downsampled CNN with one more layer than $\tilde{f}_n$ since linear map $A$ can be realized by one ReLU layer. Note that for any $x \in \mathcal{M}_\rho\backslash\Omega$ and $z = Ax$, there exists a $\tilde{x} \in \mathcal{M}$ such that $\|x - \tilde{x}\|_2 \le \rho$, and

$$|f_n^*(x) - f_0(x)| \le |\tilde{f}_n(Ax) - \tilde{F}_0(Ax)| + |\tilde{F}_0(Ax) - \tilde{F}_0(A\tilde{x})| + |\tilde{F}_0(A\tilde{x}) - f_0(x)|$$

$$\le 18\frac{B_0}{(1-\delta)^\beta}(\lfloor\beta\rfloor + 1)^2 d^{1/2} d_\delta^{(3\beta+1)/2}(NM)^{-2\beta/d_\delta} + \frac{B_0}{1-\delta}\|Ax - A\tilde{x}\|_2 + |f_0(\tilde{x}) - f_0(x)|$$

$$\le 18\frac{B_0}{(1-\delta)^\beta}(\lfloor\beta\rfloor + 1)^2 d^{1/2} d_\delta^{(3\beta+1)/2}(NM)^{-2\beta/d_\delta} + \frac{\rho B_0}{1-\delta}\sqrt{\frac{d}{d_\delta}} + \rho B_0$$

$$\le (18 + C_2)\frac{B_0}{(1-\delta)^\beta}(\lfloor\beta\rfloor + 1)^2 d^{1/2} d_\delta^{(3\beta+1)/2}(NM)^{-2\beta/d_\delta},$$

where $C_2 > 0$ is a constant not depending on any parameter and the last inequality follows from $\rho \le C_2(NM)^{-2\beta/d_\delta}(\lfloor\beta\rfloor + 1)^2 d^{1/2} d_\delta^{(3\beta+1)/2}\{\sqrt{d/d_\delta} + 1 - \delta\}^{-1}(1-\delta)^{1-\beta}$. Given the probability

measure $\nu$ of $X$ is absolutely continuous with respect to the Lebesgue measure, we have

$$\mathbb{E}|f_n^*(X) - f_0(X)| \leq (18 + C_2)\frac{B_0}{(1-\delta)^\beta}(\lfloor\beta\rfloor + 1)^2 d^{1/2} d_\delta^{(3\beta+1)/2}(NM)^{-2\beta/d_\delta}, \quad (11)$$

where $d_\delta = O(d_\mathcal{M}\log(d/\delta)/\delta^2)$ is assumed to satisfy $d_\delta \ll d$. This completes the proof of Theorem 2.4. $\qquad\square$

## 5.4  Proof of Theorem 4.2

The proof follows Theorem 11.8 and Corollary 3.19 in [21]. Recall that $(X, Y)$ has joint distribution $\mathbb{P}$, we let $\mathbb{P}_n$ denote the empirical distribution of the sample $S = \{(X_i, Y_i)\}_{i=1}^n$ and let $(\tilde{X}, \tilde{Y}) \sim \mathbb{P}_n$. Note that for any $f \in \mathcal{F}_{\mathrm{CNN}}$ we have $\|f\|_\infty \leq \mathcal{B}$, then $0 \leq \phi(yf(x)) \leq \phi_B$ for any $(x, y) \sim \mathbb{P}$. For any $f \in \mathcal{F}_{\mathrm{CNN}}$ and $t \geq 0$, we denote by $c(f, t)$ the classifier defined by $c(f, t) : (x, y) \mapsto \mathbb{1}(\phi(Yf(X)) > t)$. Correspondingly, we define the risk of $c(f, t)$ by

$$R(c(f, t)) = \mathbb{P}[c(f, t)(X, Y) = 1] = \mathbb{P}[\phi(Yf(X)) > t],$$

and similarly define its empirical risk by $R_n(c(f, t)) = \mathbb{P}_n[\phi(\tilde{Y}f(\tilde{X})) > t]$. Then we write

$$|R(f) - R_n(f)| = \left|\mathbb{E}_{(X,Y)}\phi(Yf(X)) - \mathbb{E}_{(\tilde{X},\tilde{Y})}\phi(\tilde{Y}f(\tilde{X}))\right|$$

$$= \left|\int_0^{\phi_B}\left(P[\phi(Yf(X)) > t] - P[\phi(\tilde{Y}f(\tilde{X})) > t]\right)dt\right|$$

$$\leq \phi_B \sup_{t\in[0,\phi_B]}\left|P[\phi(Yf(X)) > t] - P[\phi(\tilde{Y}f(\tilde{X})) > t]\right|$$

$$= \phi_B \sup_{t\in[0,\phi_B]}\left|R(c(f, t)) - R_n(c(f, t))\right|.$$

And this implies that

$$P\left[\sup_{f\in\mathcal{F}_{\mathrm{CNN}}}|R(f) - R_n(f)| > \epsilon\right] \leq P\left[\sup_{f\in\mathcal{F}_{\mathrm{CNN}},t\in[0,\phi_B]}|R(c(f, t)) - R_n(c(f, t))| > \frac{\epsilon}{\phi_B}\right].$$

The right-hand side can be bounded using a standard generalization bound for classification (Corollary 3.19 in [21]) in terms of the VC-dimension of the class of functions $\{c(f, t) : f \in \mathcal{F}_{\mathrm{CNN}}, t \in [0, \phi_B]\}$, which, by definition of the pseudo-dimension, is $\mathrm{Pdim}(\mathcal{F}_{\mathrm{CNN}})$ the pseudo-dimension of function class $\mathcal{F}_{\mathrm{CNN}} = \{(x, y) \mapsto \phi(yf(x)) : f \in \mathcal{F}_{\mathrm{CNN}}\}$. Now we have with probability at least $1 - \delta$ over the choice of an i.i.d. sample $S$ of size $n$ that

$$\sup_{f\in\mathcal{F}_{\mathrm{CNN}}}|R(f) - R_n(f)| \leq \phi_B\left(\sqrt{\frac{2\mathrm{Pdim}(\mathcal{F}_{\mathrm{CNN}})\log(en)}{n}} + \sqrt{\frac{\log(1/\delta)}{2n}}\right).$$

Next, we bound the pseudo-dimension of $\mathcal{F}_{\mathrm{CNN}}$ by that of $\mathcal{F}_{\mathrm{CNN}}$, and this leads to the final stochastic error bound in terms of the parameters of $\mathcal{F}_{\mathrm{CNN}}$. For a sample $S = \{(X_i, Y_i)\}_{i=1}^n$ with size $n$, if there exist $t_1, \ldots, t_n \in [0, \phi_B]$ such that for each $(b_1, \ldots, b_n) \in \{0, 1\}^n$ there exists a $f \in \mathcal{F}$ such that $\mathbb{1}(\phi(y_if(x_i)) > t_i) = b_i$ for $i = 1, \ldots, n$, then since $\phi$ is convexity and non-increasing, we also have $\mathbb{1}(f(x_i) > \phi^{-1}(t_i)) = 1 - b_i$ if $y_i = 1$ and $\mathbb{1}(f(x_i) > -\phi^{-1}(t_i)) = b_i$ if $y_i = -1$. Thus $\{x_i\}_{i=1}^n$ is shattered by $\mathcal{F}_{\mathrm{CNN}}$ and the threshold values $y_1\phi^{-1}(t_1), \ldots, y_n\phi^{-1}(t_n)$ witness the shattering. This implies $\mathrm{Pdim}(\mathcal{F}_{\mathrm{CNN}}) \leq \mathrm{Pdim}(\mathcal{F}_{\mathrm{CNN}})$. Besides, for piece-wise linear neural networks $\mathcal{F}_{\mathrm{CNN}}$, it is proved in Theorem 6 of [3] that

$$\mathrm{Pdim}(\mathcal{F}_{\mathrm{CNN}}) \leq C \cdot \mathcal{S}L\log\mathcal{S},$$

for some universal constant $C > 0$ where $\mathcal{S}$ and $L$ are the size and depth of $\mathcal{F}_{\mathrm{CNN}}$ respectively. Combine above inequities, we have with probability at least $1 - \delta$ over the choice of an i.i.d. sample $S$ of size $n$ that

$$\sup_{f\in\mathcal{F}_{\mathrm{CNN}}}|R(f) - R_n(f)| \leq \phi_B\left(\sqrt{\frac{C\mathcal{S}L\log\mathcal{S}\log(n)}{n}} + \sqrt{\frac{\log(1/\delta)}{n}}\right),$$

for some universal constant $C > 0$.

## 5.5 Proof of Lemma 4.4

*Proof.* To deal with the approximation of unbounded target function, we consider truncating the target function $f_0$ by a constant $T$, where $T$ may depend on $n$. Let $f_{0,T}$ be the truncated version of $f_0$ defined as

$$f_{0,T}(x) = \begin{cases} f_0(x), & \text{if } |f_0(x)| \leq T, \\ T\operatorname{sign}(f_0(x)), & \text{if } |f_0(x)| > T. \end{cases}$$

Then, the approximation error can be decomposed into two terms that are easier to deal with. Denote the error of the loss function $\phi$ due to truncation by

$$\Delta_\phi(T) := \phi(T) - \inf_{a \in \operatorname{Ran}(f_0)} \phi(a), \tag{12}$$

where $\operatorname{Ran}(f_0)$ is the range of the target function $f_0$. This error decreases as the threshold $T$ increases, however, a bigger $T$ leads to bigger $\mathcal{B}$ and $B_\phi$, thus a bigger stochastic error.

Firstly, we prove that under Assumption 4.3(a) and (b), $f_{0,T}$, the truncated version of the target function $f_0$, is bounded by $T$ and continuous on $\mathcal{X}$. Recall that for any $a \in \mathbb{R}$ and $\eta \in [0,1]$, the conditional risk function $H(\eta, a) = \eta\phi(a) + (1-\eta)\phi(-a)$. Given $\eta \in [0,1]$, the function $dH(\eta, a)/da = \eta\phi'(a) - (1-\eta)\phi'(-a)$ is continuous in both $\eta$ and $a$. Thus under Assumption 4.3 (b) the solution $a(\eta) = \arg\min_{a \in \mathbb{R}} H(\eta, \alpha)$ is continuous with respect to $\eta$, and $|a(\eta)| < \infty$ when $\eta \in (0,1)$. By Assumption 4.3 (a), $\eta(x)$ is continuous on $\{x \in \mathcal{X} : \eta(x) \leq 1 - \delta\}$ for any $\delta \in (0,1)$, thus $f_0(x) = f_0(\eta(x))$ is continuous and $\eta(x)\phi'(f_0(x)) - (1-\eta(x))\phi'(-f_0(x)) = 0$ on for any $x \in \mathcal{X}$. And $f_0(x)$ is continuous on $\{x \in \mathcal{X} : |f_0(\eta(x))| \leq T\}$ for any $T \geq 0$. For any $T > 0$, the set $\{x \in \mathcal{X} : |f_0(x)| \leq T\}$ is a compact set. Thus $f_{0,T}$, the truncated version of target function is also continuous on $\mathcal{X}$.

Recall that for any $f$, the $\phi$-risk is defined by $R(f) = \mathbb{E}\phi(Yf(X))$, then for $f_{0,T}$, the truncated version of target function satisfies

$$
\begin{aligned}
R(f_{0,T}) - R(f_0) &= \mathbb{E}\{\phi(Yf_{0,T}(X)) - \phi(Yf_0(X))\} \\
&= \mathbb{E}\Big[\mathbb{E}\{\phi(Yf_{0,T}(X)) - \phi(Yf_0(X)) \mid X\}\Big] \\
&= \mathbb{E}\Big[\eta(X)\{\phi(f_{0,T}(X)) - \phi(f_0(X))\} + (1 - \eta(X))\{\phi(-f_{0,T}(X)) - \phi(-f_0(X))\}\Big].
\end{aligned}
$$

For $x \in \mathcal{X}$, we define $g(x) = \eta(x)\{\phi(f_{0,T}(x)) - \phi(f_0(x))\} + (1 - \eta(x))\{\phi(-f_{0,T}(x)) - \phi(-f_0(x))\}$. Recall that $\phi$ is non-increasing and we have:

(a) If $-T \leq f_0(x) \leq T$, then $f_0(x) = f_{0,T}(x)$ and $g(x) = 0$;

(b) If $f_0(x) > T$, then $f_{0,T}(x) = T$ and $\phi(-f_{0,T}(x)) - \phi(-f_0(x)) \leq 0$, and

$$
\begin{aligned}
g(x) &= \eta(x)\{\phi(f_{0,T}(x)) - \phi(f_0(x))\} + (1 - \eta(x))\{\phi(-f_{0,T}(x)) - \phi(-f_0(x))\} \\
&\leq \eta(x)\{\phi(f_{0,T}(x)) - \phi(f_0(x))\} \\
&= \eta(x)\{\phi(T) - \phi(f_0(x))\} \\
&\leq \phi(T) - \phi(f_0(x));
\end{aligned}
$$

(c) If $f_0(x) < -T$, then $f_{0,T}(x) = -T$ and $\phi(f_{0,T}(x)) - \phi(f_0(x)) \leq 0$, and

$$
\begin{aligned}
g(x) &= \eta(x)\{\phi(f_{0,T}(x)) - \phi(f_0(x))\} + (1 - \eta(x))\{\phi(-f_{0,T}(x)) - \phi(-f_0(x))\} \\
&\leq (1 - \eta(x))\{\phi(-f_{0,T}(x)) - \phi(-f_0(x))\} \\
&= (1 - \eta(x))\{\phi(T) - \phi(-f_0(x))\} \\
&\leq \phi(T) - \phi(-f_0(x)).
\end{aligned}
$$

Thus for any $x \in \mathcal{X}$, we have $g(x) \leq \phi(T) - \inf_{a \in \operatorname{Ran}(f_0)} \phi(a)$ and $R(f_{0,T}) - R(f_0) = \mathbb{E}\{g(X)\} \leq \phi(T) - \inf_{a \in \operatorname{Ran}(f_0)} \phi(a)$, where $\operatorname{Ran}(f_0)$ is the range of the target function $f_0$.

$$\inf_{f \in \mathcal{F}_{\mathrm{CNN}}} R(f) - R(f_0) = \inf_{f \in \mathcal{F}_{\mathrm{CNN}}} R(f) - R(f_{0,T}) + R(f_{0,T}) - R(f_0),$$

then left thing to do is to prove

$$\inf_{f \in \mathcal{F}_{\text{CNN}}} R(f) - R(f_{0,T}) \leq B_\phi \inf_{f \in \mathcal{F}_{\text{CNN}}} \mathbb{E}|f(X) - f_{0,T}(X)|.$$

With $f_0 \in \mathcal{H}^\beta([0,1]^d, B_0)$, $\phi$ is $\mathcal{B}_\phi$-Lipschitz, and it is a straightforward proof that

$$
\begin{aligned}
\inf_{f \in \mathcal{F}_{\text{CNN}}} R(f) - R(f_{0,T}) &= \inf_{f \in \mathcal{F}_{\text{CNN}}} \mathbb{E}\{\phi(Yf(X)) - \phi(Yf_{0,T}(X))\} \\
&\leq \inf_{f \in \mathcal{F}_{\text{CNN}}} \mathbb{E}\{\mathcal{B}_\phi|Yf(X) - Yf_{0,T}(X)|\} \\
&= B_\phi \inf_{f \in \mathcal{F}_{\text{CNN}}} \mathbb{E}|f(X) - f_{0,T}(X)|.
\end{aligned}
$$

$\square$

## 5.6 Proof of Theorem 4.5

The main focus of the proof is to deal with the truncation. If $f_0 \in \mathcal{H}^\beta([0,1]^d, B_0)$ is uniformly bounded by $T \leq \mathcal{B}$, by Theorem 2.2, there exists a ReLU neural network $\phi$ such that

$$\mathbb{E}|\phi(X) - f_0(X)| \leq 18 B_0 (\beta+1)^2 d^{\beta + (\beta \vee 1)/2} (NM)^{-2\beta/d}.$$

We let $\phi_0(x) = \sigma(\phi(x) + \mathcal{B}) - \sigma(\phi(x) - \mathcal{B}) - \mathcal{B}$ be the truncated network where $\sigma$ denotes the ReLU activation function. Then

$$\mathbb{E}|\phi_0(X) - f_{0,T}(X)| \leq \mathbb{E}|\phi(X) - f_0(X)| \leq 18 B_0 (\beta+1)^2 d^{\beta + (\beta \vee 1)/2} (NM)^{-2\beta/d}.$$

If $f_0 \in \mathcal{H}^\beta([0,1]^d, B_0)$ is not uniformly bounded by $T$, then the set $\{x : |f_0(x)| \leq T\}$ does not equal to the whole domain $\mathcal{X}$. Actually $\{x : |f_0(x)| \leq T\}$ can be written as a union of connected subsets since $f_0$ is Hölder smooth on its domain $\mathcal{X} = [0,1]^d$, and in the interior of each connected subset, the function $f_0$ keeps its Hölder smoothness $\beta$. By [8], the truncated target function $f_{0,T}$ on $\{x \in \mathcal{X} : |f_0(x)| \leq T\}$ can be extended to $\mathcal{X}$ to be $\beta$-Hölder and bound $T + c$ for some small constant $c > 0$. Let $\tilde{f}_{0,T}$ be the extended function on $\mathcal{X}$, then by Theorem 2.2, there exists a ReLU neural network $\phi$ such that

$$\mathbb{E}|\phi(X) - \tilde{f}_{0,T}(X)| \leq 18 B_0 (\beta+1)^2 d^{\beta + (\beta \vee 1)/2} (NM)^{-2\beta/d}.$$

We let $\phi_0(x) = \sigma(\phi(x) + \mathcal{B}) - \sigma(\phi(x) - \mathcal{B}) - \mathcal{B}$ be the truncated network where $\sigma$ denotes the ReLU activation function. Then

$$\mathbb{E}|\phi_0(X) - f_{0,T}(X)| \leq \mathbb{E}|\phi(X) - \tilde{f}_{0,T}(X)| \leq 18 B_0 (\beta+1)^2 d^{\beta + (\beta \vee 1)/2} (NM)^{-2\beta/d}.$$

## 5.7 Proof of Theorem 4.6

Combining Lemma 4.1, Theorem 4.2 and 4.5, Theorem 4.6 is easily proved.

For the proof of Theorem 4.7, we need the following lemma, which can be proved using the technique in [23]. We refer to [23] for the detailed description of the technique.

## 5.8 Proof of Theorem 4.7

*Proof.* The proof is similar to that of Theorem 4.6, except how we deal with $\inf_{f \in \mathcal{F}_{\text{CNN}}} \mathbb{E}\|f(X) - f_{0,T}(X)\|_2$. Combining Lemma 4.4 and using Theorem 2.4, we have

$$
\begin{aligned}
R(\hat{f}_n) - R(f_0) &\leq \frac{16\sqrt{2} B_\phi \mathcal{B} \mathcal{S}^{1/2} L^{1/4}}{\sqrt{n}} + 2 B_\phi \mathcal{B} \sqrt{\frac{2\log(1/\delta)}{n}} \\
&\quad + (18 + C_2) \frac{B_0}{(1-\varepsilon)^\beta} (\lfloor\beta\rfloor + 1)^2 d^{1/2} d_\varepsilon^{3\lfloor\beta\rfloor/2} (NM)^{-2\beta/d_\varepsilon} + \inf_{|a| \leq T} \phi(a) - \inf_{a \in E_\phi} \phi(a).
\end{aligned}
$$

This completes the proof of Theorem 4.7. $\square$

## 5.9 Proof of the cross entropy loss example in Section 5

For the cross entropy loss function $\phi(a) = -\log\{0.5 + a\}$, with the choice $T = \mathcal{B} = 0.5 - n^{\beta/(2d+4\beta)}$, we have $\mathcal{B} \leq 0.5$, $\Delta_\phi(T) \leq n^{-\beta/(2d+4\beta)}$ and $B_\phi = n^{\beta/(2d+4\beta)}$. Suppose Assumptions 2.3, 4.3 hold and $f_0 \in \mathcal{H}^\beta([0,1]^d, B_0)$.

Take $N = 1$ in (7) and $M = \lfloor \xi \rfloor$ in (4) and let $\mathcal{F}_{\text{CNN}}$ be the class of CNNs in (2) with depth $L \leq 378 \cdot 38^2(\lfloor \beta \rfloor + 1)^6 d^{2\lfloor \beta \rfloor + 2} \lfloor n^{d/\{2d+4\beta\}} \rfloor \lceil \log(8n) \rceil / (s_{\min} - 1)$, filter lengths $2 \leq s_{\min} \leq s_{\max} \leq 9 \times 38^2(\lfloor \beta \rfloor + 1)^4 d^{2\lfloor \beta \rfloor + 2}$ and size $\mathcal{S} \leq 8\mathcal{W}L \leq 42 * 8 * 9^2 * 38^4(\lfloor \beta \rfloor + 1)^{10} d^{4\lfloor \beta \rfloor + 4} \lfloor n^{d/\{2d+4\beta\}} \rfloor \lceil \log(8n) \rceil / (s_{\min} - 1)$.

Theorem 4.6 implies that, with probability at least $1 - \exp\{-n^{d/(d+2\beta)}\}$, the excess $\phi$-risk of the ERM $\hat{f}_n$ defined in (13) satisfies

$$R(\hat{f}_n) - R(f_0) \leq n^{-\beta/(2d+4\beta)}$$
$$\times \{c(\lfloor \beta \rfloor + 1)^8 d^{3\lfloor \beta \rfloor + 3}(\log n)^2 / (s_{\min} - 1) + 18B_0(\lfloor \beta \rfloor + 1)^2 d^{\lfloor \beta \rfloor + (\lfloor \beta \rfloor \vee 1)/2}\},$$

or simply

$$R(\hat{f}_n) - R(f_0) \leq C(d, \beta, B_0, s_{\min})(\log n)^2 n^{-\beta/(2d+4\beta)},$$

where $C(d, \beta, B_0, s_{\min}) = O(B_0(\beta + 1)^8 d^{3\beta + 3}/(s_{\min} - 1))$ is a constant independent of $n$. The excess risk bound under the approximate low-dimensional manifold assumption can be obtained in a similar way.

Under the modified cross entropy loss $\phi(a) = \max\{-\log\{0.5 + a\}, \tau\}$ with $\tau = -\log\{0.5 + T\}$, the corresponding measurable minimizer defined in (14) is $f_{0,T}$ and $\Delta_\phi(T) = 0$ by definition since the infimum of the modified cross entropy loss can be achieved within $[-T, T] \subset [-0.5, 0.5]$. By choosing $T = \mathcal{B} = 0.5 - (\log n)^{-1}$ and $\tau = -\log\{1 - (\log n)^{-1}\}$, with the modified cross entropy loss, Theorem 4.6 implies that with probability at least $1 - \exp\{-n^{d/(d+2\beta)}\}$, the excess $\phi$-risk of the ERM $\hat{f}_n$ defined in (13) satisfies

$$R(\hat{f}_n) - R(f_0) \leq C(d, \beta, B_0, s_{\min})(\log n)^2 n^{-\beta/(d+2\beta)},$$

where $C(d, \beta, B_0, s_{\min}) = O(B_0(\lfloor \beta \rfloor + 1)^8 d^{3\lfloor \beta \rfloor + 3}/(s_{\min} - 1))$ is a constant independent of $n$.

## 5.10 Proof of the SVM example

*Proof.* Based on Lemma 4.1, our proof focus on controlling the approximation error $\inf_{f \in \mathcal{F}_{\text{CNN}}} R(f) - R(f_0)$. Recall that $f_0(x) = \text{sign}(2\eta(x) - 1)$, we would firstly approximate Hölder smooth function $\eta$ by CNNs. Similar to the proof in Theorem 4.6, for any $N, M \in \mathbb{N}^+$, there exists a function $\eta^* \in \mathcal{F}_{CNN}$ with bound $\mathcal{B} = 1$, depth $L \leq 378 \cdot 38^2(\lfloor \beta \rfloor + 1)^6 d^{2\lfloor \beta \rfloor + 2} M \lceil \log(8M) \rceil / (s_{\min} - 1)$, filter lengths $2 \leq s_{\min} \leq s_{\max} \leq 38^2(\lfloor \beta \rfloor + 1)^4 d^{2\lfloor \beta \rfloor + 2} N \lfloor \log(8N) \rfloor$ and size $\mathcal{S} \leq 8\mathcal{W}L \leq 42 * 8 * 38^4(\lfloor \beta \rfloor + 1)^{10} d^{4\lfloor \beta \rfloor + 4} M \lceil \log(8M) \rceil N^2 \lceil \log(8N) \rceil^2 / (s_{\min} - 1)$ such that

$$|\eta^*(x) - \eta(x)| \leq 18B_0(\lfloor \beta \rfloor + 1)^2 d^{\lfloor \beta \rfloor + (\beta \vee 1)/2}(NM)^{-2\beta/d},$$

for any $x \in \mathcal{X} \backslash \Omega$ where $\Omega$ is a set with arbitrarily small Lebesgue measure. Let $\varepsilon_n = 18B_0(\lfloor \beta \rfloor + 1)^2 d^{\lfloor \beta \rfloor + (\beta \vee 1)/2}(NM)^{-2\beta/d}$.

We construct the neural network $f_{CNN}$ by adding one layer to $\eta^*$ as

$$f_{CNN}(x) = 2\sigma\left[\frac{1}{\varepsilon_n}\{\eta^*(x) - \frac{1}{2}\}\right] - 2\sigma\left[\frac{1}{\varepsilon_n}\{\eta^*(x) - \frac{1}{2}\} - 1\right] - 1,$$

where $\sigma$ is the ReLU activation function. Then we can see that $\|f_{CNN}\|_\infty \leq 1$ and $f_{CNN}(x) = 1$ if $\eta^*(x) \geq \varepsilon_n + 1/2$, $f_{CNN}(x) = 2(\eta^*(x) - 1/2)/\varepsilon_n - 1$ if $1/2 + \varepsilon_n > \eta^*(x) \geq 1/2$ and $f_{CNN}(x) = -1$ if $\eta^*(x) < 1/2$. Let $\Omega_{\eta,\varepsilon} = \{x \in \mathcal{X} : |2\eta(x) - 1| > \varepsilon\}$ for $\epsilon > 0$. Then for $x \in \Omega_{\eta,4\varepsilon_n}$, we have $|f_{CNN}(x) - f_0(x)| = 0$, since $\eta^*(x) - 1/2 = (\eta(x) - 1/2) - (\eta^*(x) - \eta(x)) > \varepsilon_n$ when

$\eta(x) - 1/2 > 2\varepsilon_n$ and $\eta^*(x) - 1/2 < -\varepsilon_n$ when $\eta(x) - 1/2 < -2\varepsilon_n$. Then

$$
\begin{aligned}
R(f_{CNN}) - R(f_0) &= \mathbb{E}\Big[\phi(Yf_{CNN}(X)) - \phi(Yf_0(X))\Big] \\
&= \mathbb{E}\Big[|f_{CNN}(X) - f_0(X)||2\eta(X) - 1|\Big] \\
&= \mathbb{E}_{X \in \Omega_{\eta,4\varepsilon_n}}\Big[|f_{CNN}(X) - f_0(X)||2\eta(X) - 1|\Big] \\
&\leq 8\varepsilon_n P(|2\eta(X) - 1| \leq 4\varepsilon_n) \\
&\leq 8 \times 4^q \times c_{noise} \times \varepsilon_n^{q+1}.
\end{aligned}
$$

Setting the parameters in $\eta^*$ by $N = 1$ and $M = \lfloor n^{d/\{d+4\beta(q+1)\}} \rfloor$ and size (number of parameters) $\mathcal{S} \leq 8\mathcal{W}L \leq 42 * 8 * 9^2 * 38^4(\lfloor\beta\rfloor+1)^{10}d^{4\lfloor\beta\rfloor+4}\lfloor n^{d/\{d+4\beta(q+1)\}}\rfloor\lceil\log(8n)\rceil/(s_{\min}-1)$, and let $\delta = \exp(-n^{d/(d+2\beta(q+1))})$, then by Theorem 4.6, with probability at least $1 - \exp(-n^{d/(d+2\beta(q+1))})$,

$$
\begin{aligned}
R(\hat{f}_n) - R(f_0) \leq \quad &(\log n)^2 \times n^{-\beta(q+1)/\{d+2\beta(q+1)\}} \\
\times &\Big\{c(\lfloor\beta\rfloor + 1)^8 d^{3\lfloor\beta\rfloor+3} + 8 \times 4^q c_{noise}[18B_0(\lfloor\beta\rfloor + 1)^2 d^{\lfloor\beta\rfloor+(\beta\vee1)/2}]^{q+1}\Big\}.
\end{aligned}
$$

The excess risk bound under the approximate low-dimensional manifold assumption can be obtained in a similar way. □

## 5.11 Proof of the logistic loss example

The proof is similar to that of the cross entropy loss example. We omitted the proof here.

## 5.12 Proof of the exponential loss example

The proof is similar to that of the cross entropy loss example. We omitted the proof here.

## 5.13 Supporting definitions and Lemmas

For ease of reference, we collect several definitions and existing results that we used in our proofs. Firstly, we present the definition and properties of the averaged Taylor expansion in chapter 4 of [5].

**Definition 5.1** (Averaged Taylor expansion). Let $\beta \in \mathbb{N}, 1 \leq p \leq \infty$ and $f \in W^{\beta-1,p}$, and let $\Omega \in \mathbb{R}^d$, $x_0 \in \Omega$, $r > 0$, $B := B_r(x_0) = \{x \in \mathbb{R}^d : \|x - x_0\|_2 < r\}$ with its closure $\bar{B}$ compact in $\Omega$. The corresponding Taylor polynomial of order $\beta$ of $f$ averaged over $B$ is defined as

$$
T^\beta f(x) = \int_B T_y^\beta f(x)\phi(y)dy, \tag{13}
$$

where

$$
T_y^\beta f(x) = \sum_{\|\alpha\|_1 \leq \beta-1} \frac{1}{\alpha!} D^\alpha f(y)(x - y)^\alpha,
$$

and $\phi$ is an arbitrary cut-off function supported in $\bar{B}$ being infinitely differentiable, i.e., $\phi \in C^\infty(\mathbb{R}^d)$ with supp $\phi = \bar{B}$ and $\int_{\mathbb{R}^d} \phi(x)dx = 1$. For example, let

$$
\psi(x) = \begin{cases} \exp\{-(1 - (|x - x_0|/r)^2)^{-1}\}, & \text{if } |x - x_0| < r, \\ 0 & , \text{ else,} \end{cases}
$$

and let $c = \int_{\mathbb{R}^d} \psi(x)dx (c > 0)$, then $\phi(x) = \psi(x)/c$ is a cut-off function and $\|\phi\|_{L^\infty(B)} := \max|\phi(x)| \leq c(d) \cdot r^{-d}$ where $c(d) > 0$ is a constant depending only on $d$. Besides,

$$
\phi(x) = \begin{cases} \pi^{-d/2}\Gamma(d/2 + 1)r^{-d}, & \text{if } |x - x_0| < r, \\ 0 & , \text{ else,} \end{cases}
$$

is a another example of cut-off function, in which case $\phi$ puts constant weight on the ball $d$-dimensional $B$ with radius $r$.

The averaged Taylor polynomial defined above is an integral of the traditional Taylor expansion weighted by the cut-off function over a region. Following the equations (4.1.5)-(4.1.8) in [5], we show the below lemma that the averaged Taylor polynomial of order $\beta$ is indeed a polynomial of degree less than $\beta$ in $x$.

**Lemma 5.2.** *Let $\beta \in \mathbb{N}, 1 \le p \le \infty$ and $f \in W^{\beta-1,p}(\Omega)$, and let $x_0 \in \Omega, r > 0, R \ge 1$ such that for the ball $B := B_r(x_0)$ with its closure $\bar{B}$ compact in $\Omega$ and $\|y\|_\infty \le R$ for all $y \in B$. Then the Taylor polynomial of order $\beta$ of $f$ averaged over $B$ can be written as*

$$T^\beta f(x) = \sum_{\|\alpha\|_1 \le \beta-1} c_\alpha x^\alpha \qquad \text{for } x \in \Omega,$$

*where*

$$|c_\alpha| \le \beta(2d)^{\beta-1}(\pi^{-d/2}\Gamma(d/2+1))^{1/p}R^{\beta-1}\|f\|_{W^{\beta-1,p}(\Omega)}r^{-d/p}$$

*for all $\alpha$ satisfying $\|\alpha\|_1 \le \beta - 1$.*

*Proof.* Recall that for $x, y \in \mathbb{R}^d$ and $\alpha = (\alpha_1, \ldots, \alpha_d) \in \mathbb{N}_0^d$, if we write

$$(x - y)^\alpha = \Pi_{i=1}^d (x_i - y_i)^{\alpha_i} = \sum_{\gamma,\theta \in \mathbb{N}_0^d, \gamma+\theta=\alpha} a(\gamma,\theta)x^\gamma y^\theta,$$

where $\gamma = (\gamma_1, \ldots, \gamma_d)$ and $\theta = (\theta_1, \ldots, \theta_d)$ are $d$-tuples of nonnegative integers, and $a(\gamma, \theta) \in \mathbb{R}$ are the constants satisfying $|a(\gamma, \theta)| \le \binom{\alpha}{\gamma} = \alpha!/(\gamma!\theta!)$ in multi-index notation. Then

$$T^\beta f(x) = \int_B T_y^\beta f(x)\phi(y)dy = \sum_{\|\alpha\|_1 \le \beta-1} \sum_{\gamma+\theta=\alpha} \frac{1}{\alpha!}a(\gamma,\theta)x^\gamma \int_B D^\alpha f(y)y^\theta\phi(y)dy$$

$$= \sum_{\|\gamma\|_1 \le \beta-1} x^\gamma \left\{ \sum_{\|\gamma+\theta\|_1 \le \beta-1} \frac{1}{(\gamma+\theta)!}a(\gamma,\theta)\int_B D^{\gamma+\theta}f(y)y^\theta\phi(y)dy \right\}.$$

Then we consider bounding the coefficient of $x^\gamma$. Note that

$$\left| \int_B D^{\gamma+\theta}f(y)y^\theta\phi(y)dy \right| \le \int_B |D^{\gamma+\theta}f(y)||y^\theta||\phi(y)|dy$$

$$\le R^{\|\theta\|_1}\|f\|_{W^{\beta-1,p}(B)}\|\phi\|_{L^q(B)}$$

$$\le R^{\|\theta\|_1}\|f\|_{W^{\beta-1,p}(B)}\|\phi\|_{L^1(B)}^{1/q}\|\phi\|_{L^\infty(B)}^{1-1/q}$$

$$\le R^{\beta-1}\|f\|_{W^{\beta-1,p}(\Omega)}\|\phi\|_{L^\infty(B)}^{1-1/q}$$

$$\le R^{\beta-1}\|f\|_{W^{\beta-1,p}(\Omega)}(\pi^{-d/2}\Gamma(d/2+1))^{1/p}r^{-d/p},$$

where the second inequality holds since $\|y\|_\infty \le R, \forall y \in B$ and the use of Hölder's inequality with $1/p + 1/q = 1$, the third inequality holds again by Hölder's inequality, and the last two lines hold since $\|\phi\|_{L^1(B)} = 1$ and $\|\phi(x)\|_{L^\infty(B)} \le \pi^{-d/2}\Gamma(d/2+1) \cdot r^{-d}$. Lastly, the coefficient $c_\gamma$ of $x^\gamma$ can be upper bounded by

$$|c_\gamma| = \left| \sum_{\|\gamma+\theta\|_1 \le \beta-1} \frac{1}{(\gamma+\theta)!}a(\gamma,\theta)\int_B D^{\gamma+\theta}f(y)y^\theta\phi(y)dy \right|$$

$$\le R^{\beta-1}\|f\|_{W^{\beta-1,p}(\Omega)}\|\phi\|_{L^\infty(B)} \sum_{\|\gamma+\theta\|_1 \le \beta-1} \frac{1}{(\gamma+\theta)!}|a(\gamma,\theta)|$$

$$\le (\pi^{-d/2}\Gamma(d/2+1))^{1/p}R^{\beta-1}\|f\|_{W^{\beta-1,p}(\Omega)}r^{-d/p} \sum_{\|\gamma+\theta\|_1 \le \beta-1} \frac{1}{\gamma!\theta!}.$$

Note that

$$\sum_{\|\gamma+\theta\|_1 \le \beta-1} \frac{1}{\gamma!\theta!} = \sum_{s=0}^{\beta-1} \sum_{\|\alpha\|_1=s} \sum_{\gamma+\theta=\alpha} \frac{1}{\gamma!\theta!}$$

$$\le \sum_{s=0}^{\beta-1} \sum_{\|\alpha\|_1=s} 2^s \le \sum_{s=0}^{\beta-1} d^s 2^s \le \beta(2d)^{\beta-1}.$$

Thus we have

$$|c_\gamma| \le \beta(2d)^{\beta-1}(\pi^{-d/2}\Gamma(d/2+1))^{1/p}R^{\beta-1}\|f\|_{W^{\beta-1,p}(\Omega)}r^{-d/p}.$$

$\qquad\qquad\qquad\qquad\qquad\qquad\qquad\qquad\qquad\qquad\qquad\qquad\qquad\qquad\qquad\qquad\square$

Next, we present the main approximation result of averaged Taylor expansion on Sobolev functions by Bramble-Hilbert (Lemma 4.3.8 in [5]) after introduce some prerequisite definitions and conceptions.

**Definition 5.3** (Star-shaped set). Let $\Omega, B \subset \mathbb{R}^d$. $\Omega$ is star-shaped with respect to $B$ if for all $x \in \Omega$, the closed convex hull of $\{x\} \cup B$ is a subset of $\Omega$.

**Definition 5.4** (Chunkiness parameter). Suppose $\Omega \subset \mathbb{R}^d$ has diameter $r_\Omega$ and is star-shaped with respect to a ball $B$. Let $r_{\max} = \sup\{r \in \mathbb{R}^+ : \Omega$ is star-shaped with respect to a ball of radius $r\}$. Then the chunkiness parameter of $\Omega$ is defined by $\gamma = r_\Omega/r_{\max}$.

**Lemma 5.5** (Proposition 4.2.8 in [5]). *Let $f \in W^{\beta,p}(\Omega)$ and $T^\beta$ be its averaged Taylor expansion defined in (13) over the ball $B$ with radius $r$, then the remainder $R^\beta f := f - T^\beta f$ satisfies*

$$R^\beta f(x) = \beta \sum_{|\alpha|=\beta} \int_{C_x} k_\alpha(x,z)D^\alpha f(x)dz,$$

*where $C_x$ is the convex hull of $\{x\} \cup B$, $z = x + s(y-x)$, $k_\alpha(x,z) = (1/\alpha!)(x-z)^\alpha k(x,z)$ and*

$$|k(x,z)| \le \frac{1}{d}\|\phi\|_{L^\infty(B)}(r+|x-x_0|)^d|z-x|^{-d}.$$

**Lemma 5.6** (Bramble-Hilbert). *Let $B$ be a ball in $\Omega$ such that $\Omega$ is star-shaped with respect to $B$ and such that its radius $r > r_{\max}/2$, where $r_{\max}$ is defined in Definition 5.4. Let $T^\beta f$ be the Taylor polynomial of order $\beta$ of $f$ averaged over $B$ where $f \in W^{\beta,p}(\Omega)$ and $p \ge 1$. Then*

$$\|f - T^\beta f\|_{W^{k,p}(\Omega)} \le \beta d^{\beta-1}\Gamma(d/2+1)\pi^{-d/2}(1+r_\Omega/r)^d r_\Omega^{\beta-k}\|f\|_{W^{k,p}(\Omega)}, \qquad k = 0,1,\ldots,\beta$$

*where $r_\Omega$ is the diameter of $\Omega$.*

The proof can be found in Lemma 4.3.8 in [5].

**Lemma 5.7** (Proposition 4.3 in [19]). *For any $N, M, d \in \mathbb{N}^+$ and $\delta \in (0, 3K]$ with $K = \lfloor N^{1/d}\rfloor^2\lfloor M^{2/d}\rfloor$, there exists a one-dimensional function $\phi$ implemented by a ReLU FNN with width $4\lfloor N^{1/d}\rfloor + 3$ and depth $4M + 5$ such that*

$$\phi(x) = k, \quad \text{if } x \in [\frac{k}{K}, \frac{k+1}{K} - \delta \cdot 1_{k<K-1}], \text{ for } k = 0,1,\ldots,K-1.$$

**Lemma 5.8** (Proposition 4.4 in [19]). *Given any $N, M, s \in \mathbb{N}^+$ and $\xi_i \in [0,1]$ for $i = 0,1,\ldots,N^2L^2 - 1$, there exists a function $\phi$ implemented by a ReLU FNN with width $16s(N+1)\lceil\log_2(8N)\rceil$ and depth $(5M+2)\lceil\log_2(4M)\rceil$ such that*

$$|\phi(i) - \xi_i| \le N^{-2s}M^{-2s}, \text{ for } i = 0,1,\ldots,N^2M^2 - 1,$$

*and $0 \le \phi(x) \le 1$ for any $x \in \mathbb{R}$.*

The next lemma demonstrate that the production function and polynomials can be approximated by ReLU neural networks. The basic idea is firstly to approximate the square function using "sawtooth" functions then the production function, which is firstly raised in [27]. A general polynomial can be further approximated combining the approximated square function and production function. The following two lemmas are more general results than those in [27].

**Lemma 5.9** (Lemma 3.4 in [14]). *For any $N, M \in \mathbb{N}^+$, and $a, b \in \mathbb{R}$ with $a < b$, there exists a function $\phi$ implemented by a ReLU FNN with width $9N + 1$ and depth $2M$ such that $\|\phi\|_{W^{1,\infty}((a,b)^2)} \le 12(b-a)^2$ and*

$$\|\phi(x,y) - xy\|_{W^{1,\infty}((a,b)^2)} \le 6(b-a)^2N^{-M}$$

*for any $x, y \in (a,b)$.*

**Lemma 5.10** (Proposition 4.1 in [19], Proposition 3.6. in [14]). *Assume $P(x) = x^\alpha = x_1^{\alpha_1}x_2^{\alpha_2}\cdots x_d^{\alpha_d}$ for $\alpha \in \mathbb{N}^d$ with $\|\alpha\|_1 \le k \in \mathbb{N}^+$. For any $N, M \in \mathbb{N}^+$, there exists a function $\phi$ implemented by a ReLU FNN with width $9(N+1) + k - 1$ and depth $14k^2M$ such that $\|\phi\|_{W^{1,\infty}(\mathcal{X})} \le 18$ and*

$$\|\phi(x) - P(x)\|_{W^{1,\infty}(\mathcal{X})} \le 10k(N+1)^{-7kM} \qquad \text{for any } x \in [0,1]^d.$$