# OpenReview forum: "Approximation with CNNs in Sobolev Space: with Applications to Classification"
_NeurIPS.cc/2022/Conference — NeurIPS 2022 Accept_

### Official Review · Reviewer_aFh9 · 2022-07-09

**Rating:** 9
**Confidence:** 5
**Soundness:** 4 excellent
**Presentation:** 4 excellent
**Contribution:** 4 excellent

**Summary:**

The authors consider deep convolutional neural networks (CNNs) for classification. Their main results include rates of approximating Sobolev smooth functions by CNNs in L_p spaces (Theorems 2.1-2.4 and 4.5), sample error bounds (Theorem 4.2), and estimates for the excess misclassification error of the induced classification algorithm (Theorems 4.6 and 4.7).

**Questions:**

Some questions have been asked in the previous section.

**Limitations:**

Yes, they admit in Assumption 4.3 that they require the loss \phi to be C^1, hence the hinge loss is not included.

**Strengths And Weaknesses:**

CNNs are the most important family of structured deep neural networks used in practice which are particularly efficient for image classification. Because of the convolutional structure, their analysis is different from that for fully-connected networks. Recently there are some results on CNNs for classification such as those of Kohler et.al. and Feng-Huang-Zhou (Generalization Analysis of CNNs
for Classification on Spheres, IEEE TNNLS, in press).

What is nice in this paper under review is the treatment of the approximation error when the target function f_0 is unbounded or not Lipschitz. The authors use a truncation operator to estimate the approximation error.

Another interesting idea of the paper is to consider an almost manifold structure of the input space and to derive learning rates in Theorem 4.7 which depend on the ambient dimension weakly. The authors should comment on the absolute continuity of the marginal distribution with respect to the Lebesgue measure.

In the proof of Theorem 4.5, the set \{x: |f_{0, T}(x)| \leq T\} is obviously the whole input space, which should be a typo. The author might mean the set \{x: |f_{0}(x)| \leq T\}. Please comment on the Hoelder smoothness of f_0 on this subset.

The parameter {\mathcal W} defined in Equation (4) should be given a network meaning. Does it stand for the maximum network width?

For the comparison theorem in terms of \psi mentioned on page 6, a simple form valid for almost all loss functions is one with \psi(u)=u^2 given by Chen-Wu-Ying-Zhou (Support vector machine soft margin classifiers: Error analysis, JMLR 5 (2004), 1143–1175).

The authors have compared their results on approximation error with the existing ones. They should do the same for the excess misclassification error.

Overall, this is an excellent paper.

---

> ### Author Response · Authors · 2022-07-31
> **Response to Reviewer aFh9**
>
> We are very grateful to you for your taking the time to read our paper and for your helpful comments and constructive suggestions.
>
> “CNNs are the most important family of structured deep neural networks used in practice which are particularly efficient for image classification. Because of the convolutional structure, their analysis is different from that for fully-connected networks. Recently there are some results on CNNs for classification such as those of Kohler et.al. and Feng-Huang-Zhou (Generalization Analysis of CNNs for Classification on Spheres, IEEE TNNLS, in press).”
>
> Response: Thank you for bringing these two articles to our attention. Indeed, they are very relevant and we have added them in the citation in the revision. We have also added additional comments in the related work part in section 1.1 in the rebuttal revision.
>
> “What is nice in this paper under review is the treatment of the approximation error when the target function f_0 is unbounded or not Lipschitz. The authors use a truncation operator to estimate the approximation error.
> Another interesting idea of the paper is to consider an almost manifold structure of the input space and to derive learning rates in Theorem 4.7 which depend on the ambient dimension weakly. The authors should comment on the absolute continuity of the marginal distribution with respect to the Lebesgue measure.”
>
> Response: Thank you very much for noticing the merits of our paper and for your suggestion on commenting the absolute continuity of the marginal distribution. Following your suggestion, we have added remarks on this point after Assumption 4.3. in the rebuttal revision. The assumption of absolute continuity of the marginal distribution of $X$ with respect to the Lebesgue measure is incompatible with the exact low-dimensional manifold assumption since the exact low-dimensional manifold has zero Lebesgue measure. In our approximate low-dimensional manifold case, the absolute continuity assumption of the marginal distribution of $X$ is reasonable. This assumption is for deriving a better $L_p$ approximation error bound, without which the error bound can still be obtained (with respect to the $L_\infty$ norm) at the price of a wider neural network and a larger prefactor.
>
>
> Question 1. “In the proof of Theorem 4.5, the set {x: |f_{0, T}(x)| \leq T} is obviously the whole input space, which should be a typo. The author might mean the set {x: |f_{0}(x)| \leq T}. Please comment on the Holder smoothness of f_0 on this subset.”
>
> Response: Thank you for pointing out this typo; we have corrected it. Also, we have commented on the Hölder smoothness of $f_0$ on the subset {$x: |f_{0}(x)| \leq T$}  in the proof of Theorem 5 in the rebuttal revision. The set {$x: |f_{0}(x)| \leq T$} can be written as a union of connected subsets since $f_0$ is Holder smooth on its domain $ [0,1]^d,$ on interior of each connected subset, the function $f_0$ keeps its Hölder smoothness. And the function $f_0$ defined on the set {$x: |f_{0}(x)| \leq T$} can be extended to the whole domain such that it keeps its Hölder smoothness and is bounded by $T+\delta$ for some tiny $\delta>0.$
>
> Question 2.
> “The parameter {\mathcal W} defined in Equation (4) should be given a network meaning. Does it stand for the maximum network width?”
> Response: Thank you for the comment. We have revised the description of parameter ${\mathcal W}$ in the rebuttal revision. In terms of the network architecture, $\mathcal{W}$ denotes the maximum incremental width (number of neurons) for consecutive layers in the network.
>
> Question 3.
> “For the comparison theorem in terms of \psi mentioned on page 6, a simple form valid for almost all loss functions is one with \psi(u)=u^2 given by Chen-Wu-Ying-Zhou (Support vector machine soft margin classifiers: Error analysis, JMLR 5 (2004), 1143–1175).
>
> Response: Thank you very much for your insightful comments. We have added comments on the general result in Theorem 34 of Chen, Wu, Ying, & Zhou (2004) on page 6, and updated the $\psi$ functions in the tables on page 6 and Appendix A in the rebuttal revision. This result shows that if the surrogate convex loss $\phi$ satisfies $\phi^{\prime\prime}(0)>0$ exists, $\phi^{’}(0)<0,$ then the corresponding $\psi(u)=u^2,$ which means the excess misclassification risk of the induced classifier can be upper bounded by the square root of the excess risk of the empirical risk minimizer under the surrogate loss.
>
> ``The authors have compared their results on approximation error with the existing ones. They should do the same for the excess misclassification error.”
>
> Response: Thank you for your constructive suggestions. Misclassification error bounds for classifiers using DNN and CNN have been obtained by Kim, Ohn and Kim (2021) and Feng, Huang and Zhou (2021), respectively. We have added comparison for the excess misclassification errors with the results from these works  in Appendix C.5 in the rebuttal revision.

---

> > ### Comment · Reviewer_aFh9 · 2022-08-08
> > **The results presented in the paper are excellent.**
> >
> > Thanks for your feedback. The rates of excess misclassification error are very nice.

---

> > > ### Author Response · Authors · 2022-08-08
> > > **Thank you very much for your positive feedback**
> > >
> > > Thank you so much for your positive feedback and encouragement. We appreciate it very much!
> > > Also, we are very grateful to you for your detailed comments in your review, which have helped us improve the paper significantly.

---

### Official Review · Reviewer_ZFjR · 2022-07-09

**Rating:** 7
**Confidence:** 2
**Soundness:** 4 excellent
**Presentation:** 3 good
**Contribution:** 3 good

**Summary:**

This paper shows new error bounds for approximating functions in Sobolev spaces with deep convolutional neural networks. The estimates depend on the dimension of the space in a polynomial way only, and explicitly show the relation to the depth and filter length of the network. By assuming that the domain of the function is well-approximated by a low-dimensional manifold, the authors can improve the error bounds significantly and thus address the curse of dimensionality. This result is exploited for deriving an error bound on the excess risk in classification tasks by separately bounding the stochastic and the approximation error, and subsequently exemplified by making the bound more concrete for the cross-entropy loss.

**Questions:**

- Please address whether a toy example for comparing theoretical and practical bounds would be feasible.
- Please briefly motivate the consideration of Sobolev functions (and/or their remaining gap to common CNN applications).

**Limitations:**

The authors have suitably addressed limitations of their work. I do not think a potential negative societal impact applies for this work.

**Strengths And Weaknesses:**

Without being an expert on the topic, this appears to be a nice contribution to me. Despite the technical nature of the paper, it is well-written and the authors make their results more accessible to non-experts by clearly defining all mathematical objects (even more standard one like Sobolev spaces). Furthermore, I like the structure of explaining the general bounds first and utilizing/exemplifying them in increasingly concrete applications (excess risk in classification; then cross-entropy in particular + more losses in the appendix). Finally, not assuming a strict manifold structure of the domain, but rather an approximate one is a strength as this assumption is surely more realistic.

Due to my limited background in the area I cannot judge how novel/significant the contribution is in comparison to prior work. From a practical perspective, I would have liked to see a toy example for some (artificial) problem on which the theoretical function approximation bounds could have been compared to numerical ones in order to get an impression of how tight or loose such estimates are. A second thing I am mostly just wondering about is if the 'approximate low dimensional manifold' assumption and the generic space of Sobolev functions is enough to capture why CNNs work well? The theoretical results in the paper partly say 'yes' (in the sense of the polynomial behavior), but practitioners often argue for CNNs if some (approximate) type of spatial invariance makes sense for the particular application. This could suggest that a much more restricted setting allow to still improve the bounds significantly (which is also the reason why I'd still expect a large gap between the derived theory and many practical applications of CNNs).

Typos: "choiceof" (without space) in equation (2); "be th number" two lines below (2).

---

> ### Author Response · Authors · 2022-07-31
> **Response to Reviewer ZFjR**
>
> We are very grateful to you for your taking the time to read our paper and for your helpful comments and constructive suggestions.
>
> “...this appears to be a nice contribution to me. ..., not assuming a strict manifold structure of the domain, but rather an approximate one is a strength as this assumption is surely more realistic.”
>
> Response: Thank you very much for noticing the merits of our paper.
>
> “...... I cannot judge how novel/significant the contribution is in comparison to prior work.”
>
> Response: There are several novel and significant aspects of our work, including better error bounds of CNN approximation in a Sobolev space, allowing the target function to be unbounded or non-Lipschitz; better prefactors in the error bounds in the sense that they depend on the ambient dimension polynomially instead of exponentially;  an explicit description of how the error bounds depend on the network architecture; and an approximation result that mitigates the curse of dimensionality when the input space is an approximate lower-dimensional manifold. As an illustration of our results, we also obtained a better learning rate for the important and basic problem of binary classification when the input space is an approximate lower-dimensional manifold.
>
> We have corrected the typos.
>
>
> Question 1.
>
> “... I would have liked to see a toy example for some (artificial) problem on which the theoretical function approximation bounds could have been compared to numerical ones.... .”
>
> “Please address whether a toy example for comparing theoretical and practical bounds would be feasible.”
>
> Response:  We have added a toy example comparing the theoretical and numerical bounds for the function approximation in Appendix D of the rebuttal revision.  Our numerical results support the theory in the sense that the approximation error shrinks by a proper rate with respect to the filter length and depth of CNN. Details are included in the table and figures in Appendix D of the rebuttal revision. The code is included in supplementary materials.
>
> Question 2.
>
> “A second thing I am mostly just wondering about is if the 'approximate low dimensional manifold' assumption and the generic space of Sobolev functions is enough to capture why CNNs work well? ....  if some (approximate) type of spatial invariance makes sense.... This could suggest that a much more restricted setting allow to still improve the bounds significantly....”
>
> "Please briefly motivate the consideration of Sobolev functions (and/or their remaining gap to common CNN applications)."
>
> Response: Our work only partially explains the empirical successes of CNNs in practice but does not fully explain why CNNs work so well in many problems. Indeed, for some image data, the assumption of approximate lower-dimensional support does not capture all the structural information. As you rightly pointed out, for example, certain types of spatial invariance properties are likely to be expected for some problems such as image classification. We agree with you and believe that if such properties are taken into account, the theoretical bounds can be further improved. This is a very interesting problem. We have added it as a limitation of our work and hope to study it in the future.
>
> “Please briefly motivate the consideration of Sobolev functions (and/or their remaining gap to common CNN applications).”
>
> Response: We used the Sobolev and Hölder classes since this is a well-established formulation to describe smoothness. Earlier works on approximation theory of neural networks have been developed for Sobolev functions, see, e.g., Yarotsky (2017); Zhou (2020); Hon & Yang (2021), and for Hölder functions, see, e.g.,  Kohler & Langer (2020); Schmidt-Hieber (2020), among others. Indeed, Sobolev and Holder classes do not capture all the characteristics of the functions encountered in some applications involving CNNs. It would be interesting to consider certain structures, such as spatial invariance as you pointed out and mentioned above, in more specifically formulated problems involving CNNs. We have included additional comments on this point in the rebuttal revision in Section 2.2 and in the description of the limitations of our work.
>
> References
>
> [1] Hon, S., & Yang, H. (2021). Simultaneous neural network approximations in Sobolev spaces. arXiv preprint arXiv:2109.00161.
>
> [2] Kohler, M., & Langer, S. (2020). Statistical theory for image classification using deep convolutional neural networks with cross-entropy loss. arXiv preprint arXiv:2011.13602.
>
> [3] Schmidt-Hieber, J. (2020). Nonparametric regression using deep neural networks with ReLU activation function. The Annals of Statistics, 48(4), 1875-1897.
>
> [4] Yarotsky, D. (2017). Error bounds for approximations with deep ReLU networks. Neural Networks, 94, 103-114.
>
> [5] Zhou, D. X. (2020). Universality of deep convolutional neural networks. Applied and computational harmonic analysis, 48(2), 787-794.

---

> > ### Comment · Reviewer_ZFjR · 2022-08-09
> > **Keeping my score**
> >
> > I'd like to thank the authors for their detailed answers, the additional numerical experiment and the convincing explanations. I am keeping my score.

---

> > > ### Author Response · Authors · 2022-08-09
> > > **Thank you for reviewing our work**
> > >
> > > Thank you so much for your positive feedback on our response and additional numerical experiment.
> > >
> > > We are very grateful to you for your work reviewing our paper and really appreciate your
> > > helpful comments and constructive suggestions that helped us improve and strengthen our paper.

---

### Official Review · Reviewer_QP4m · 2022-07-10

**Rating:** 4
**Confidence:** 5
**Soundness:** 3 good
**Presentation:** 3 good
**Contribution:** 2 fair

**Summary:**

In this paper the authors have constructed new error bounds for approximating functions in Sobolev and Holder space using deep CNNs. While such bounds have been constructed before, the authors show that there is more flexibility in decreasing these bounds by controlling network depth and filter size in comparison to previous studies. Furthermore, the new error bounds have been shown to depend on the functions’ ambient space dimension polynomially instead of exponentially. In this way the new bounds are much tighter. They also show that if the target function is approximately supported over a lower dimensional manifold, then the tightness of the error bound is further increased phenomenally, thus mitigating the curse of dimensionality and providing faster convergence.
Next, they apply the new approximation error bounds to error analysis of binary classification tasks using deep CNNs. If Y is the target binary label, and f(X) is the function modeled by a deep CNN to estimate Y, the task is to find an appropriate function from the functional space where f(X)  minimizes the expected value of some convex loss function of Y and f(X). As relevant probability distributions over f(X) and Y are unknown  beforehand, an empirical version of the misclassification risk is minimized. The authors show that the difference between the expected misclassification risk and the empirical one is lower bounded by a sum of a stochastic error and an approximation error. They further calculate upper bounds on these two error terms as well.
The stochastic error depends on the complexity of the space of f(X). Sobolev space ensures some degree of smoothness in this case. The approximation error depends on the approximating power of f(X). Furthermore, they show that the bounds on stochastic error don’t require the parameters of the CNN to be uniformly bounded. Earlier studies have enforced such constraints. Thus, the authors’ bounds aid in providing more flexibility over the range of parameter values. Then, the authors show that the bound on approximation error can be further controlled by the controlling overall bound of f(X) itself. The bound for the stochastic error increases with the size and the depth of CNN while that of approximation error decreases.
Finally in their detailed appendix, the authors construct these bounds explicitly for common loss functions for binary classification namely, least squares, hinge, logistic and exponential loss.


**Questions:**

1. Why did not the authors include more real functional approximation/regress including denoising and such where smoothness issues are front and center  rather than binary function cases?


**Ethics Review Area:**

["I don’t know"]

**Limitations:**

As noted in the last rubric, the authors’ derivations depend on binary misclassification risk. However, CNNs may have numerous objectives including multivariate classification and regression tasks. The authors admit these limitations. While deriving bounds for multivariate classifications could be straightforward, it would be far more important  to see  similar tighter bounds  for regression tasks in general.

**Strengths And Weaknesses:**

The work is original and provides a  good set of  references
The authors have provided detailed substantiation of the results  presented in the paper. They have also provided detailed calculations of the relevant bounds for common loss functions used in binary classification tasks.

Quality: The paper appears sound and is clear in stating the objectives and results.
The work is of potential  significance.
The bounds presented apply to a large class of deep CNN’s used for binary classification.
However, the work is presented only for binary classification tasks, and the authors appear to have overlooked some recent work on exactly the same topic  H. Jiang, Z. Li  and Q. Li and P. Combettes and J.C. Pesquet.

---

> ### Author Response · Authors · 2022-07-31
> **Response to Reviewer QP4m**
>
> We are very grateful to you for your taking the time to read our paper and for your helpful comments and constructive suggestions.
>
> “Quality: The paper appears sound and is clear in stating the objectives and results. The work is of potential significance. The bounds presented apply to a large class of deep CNNs used for binary classification.”
>
> Response: Thank you very much for noticing the merits of our paper.
>
> “However, the work is presented only for binary classification tasks, and the authors appear to have overlooked some recent work on exactly the same topic H. Jiang, Z. Li and Q. Li and P. Combettes and J.C. Pesquet.”
>
> Response: The main results of our paper are on CNN approximation to functions in a Sobolev space. Indeed, we have only applied our main results to binary classification, which is a basic important problem in machine learning. However, our main results are applicable to other problems that use CNNs approximations. Please also see our comments below about the general applicability of our main results, including a brief description of the application to regression.
>
> Thank you for bringing these two references, H. Jiang, Z. Li and Q. Li and P. Combettes and J.C. Pesquet, to our attention.  Since you did not provide the details of these two references, we did an extensive search and guessed that H. Jiang, Z. Li and Q. Li you were referring to is Jiang, Li and Li (2021)[1]. This work focused on the approximation properties of convolutional architectures for the target functions defined in infinite time domains tailored for temporal sequence data. Such target functions are different from those in a Sobolev space on $R^d$ in our paper. We have added Jiang, Li and Li (2021) in the citation in the revision.
>
> For the work by P. Combettes and J.C. Pesquet you mentioned, after an extensive search, we guessed you were referring to Combettes and Pesquet (2020) [2]. However, this paper is not directly related to our work. Indeed, Combettes and Pesquet (2020) focus on obtaining Lipschitz constants for feed-forward neural networks, which concerns the robustness of feed-forward neural networks in the presence of perturbations of the inputs.  Our paper studies a different problem, i.e., approximation properties of CNNs.
> However, please let us know if this is not the work by P. Combettes and J. C. Pesquet you referred to, we would be happy to read the correct reference.
>
> Question 1.
> “Why did not the authors include more real functional approximation/regress including denoising ......?”
>
> “As noted in the last rubric, the authors’ derivations depend on binary misclassification risk. ...... it would be far more important to see similar tighter bounds for regression tasks in general.”
>
> Response: We would like to mention that our main results including Theorems 2.1, 2.2, and 2.4 do not depend on a binary classification risk, instead, they concern approximation properties of CNNs for functions in a Sobolev space. We then apply our main results to binary classification problems. We use the binary classification problems to illustrate the applications of our main results since these are important basic problems and CNNs are commonly used in image classification. We appreciate your comments about the applications of our results to other settings such as regression. We have included additional remarks regarding the applications of our main results to other problems in Section 1.2, item (ii) in the rebuttal revision.
>
> To briefly illustrate the point that our results are applicable to problems involving the use of CNNs in general, consider a nonparametric regression model, $Y=f_0(X)+e,$ where $f_0$ is the unknown target function, $X$ is a $d$-dimensional predictor, $Y$ is a response variable and e is the noise independent of $X.$ Suppose $f_0$ is in a Sobolev space with smoothness index $\beta$ and we use CNNs to approximate $f_0$. Then using the same technique as in our paper, we can obtain error bounds $E [\hat{f}(X)-f_0(X)]^2\le c_1 WL^2/n +c_2 (WL)^{-4\beta/d},$ where $\hat{f}$ is the least square empirical risk minimizer over the class of CNNs, $n$ is the sample size, $W$ and $L$ denote the filter length and network depth, respectively. Under a proper choice for $W$ and $L$ with respect to the sample size n, the bounds  $E [\hat{f}(X)-f_0(X)]^2$ can achieve the nonparametric minimax optimal rate $n^{-2\beta/(2\beta+d)}$.
>
>
> References
>
> [1] Jiang, H., Li, Z., & Li, Q. (2021). Approximation Theory of Convolutional Architectures for Time Series Modelling. In International Conference on Machine Learning (pp. 4961-4970). PMLR.
>
>
> [2] Combettes, P. L., & Pesquet, J. C. (2020). Lipschitz certificates for layered network structures driven by averaged activation operators. SIAM Journal on Mathematics of Data Science, 2(2), 529-557.

---

> > ### Comment · Reviewer_QP4m · 2022-08-06
> > **Read**
> >
> > I am ok with the responses but I do take issue about the "Lipshitz certificates..." paper by PC and JCP.
> > When read carefully, the paper provides the theoretical conditions for a NN to work in some sense, the Functional properties to keep things stable and performing...

---

> > > ### Author Response · Authors · 2022-08-06
> > > **About the paper Combettes and Pesquet (2020) you suggested**
> > >
> > > Thank you for your additional comments. Indeed, as we stated in our response,
> > > "Combettes and Pesquet (2020) focus on obtaining Lipschitz constants for feed-forward neural networks, which concerns the robustness of feed-forward neural networks in the presence of perturbations of the inputs."
> > > So, we also think that controlling the Lipschitz constant is helpful with regard to the robustness of neural network models.  Therefore, we actually agree with you on the point that Combettes and Pesquet (2020) and our paper are broadly related in some tangential sense since they both deal with some aspects of neural networks models, although our work focuses on the approximation properties of CNNs, which is a different topic from the robustness and stability property of a feed-forward network. Thank you again for helping us understanding your point better, we really appreciate your help!

---

> > > ### Author Response · Authors · 2022-08-09
> > > **About Combettes and Pesquet (2020)**
> > >
> > > Thank you very much again for your work reviewing our paper.
> > >
> > > We are just wondering if you have had a chance to
> > > take a look at our response to your additional comments on Combettes and Pesquet (2020) you mentioned in your review:
> > >
> > > `` "....the authors appear to have overlooked some recent work on exactly the same topic H. Jiang, Z. Li and Q. Li and P. Combettes and J.C. Pesquet." ``
> > >
> > > Just to make sure we did not misunderstand you, we went back to read Combettes and Pesquet (2020) again. It is indeed on controlling the Lipschitz constant and related to the robustness of neural network modeling. But our work focuses on the approximation properties of CNNs in a Sobolev space.
> > >
> > > So again, we agree with you that in a broad sense, Combettes and Pesquet (2020) is somewhat related. However, we feel that our paper is not
> > >
> > > ``"...on exactly the same topic ..."`` as P. Combettes and J.C. Pesquet (2020)
> > >
> > > as you stated in your review.
> > > We would be happy to provide additional clarification on this point.
> > >
> > > Thank you so much once more for your taking the time to review our work.
> > >
> > > PS. For your convenience, we include the link to Combettes and Pesquet (2020), the title and the abstract of this paper below:
> > >
> > > https://epubs.siam.org/doi/abs/10.1137/19M1272780?journalCode=sjmdaq
> > >
> > > Patrick L. Combettes and Jean-Christophe Pesquet (2020).  Lipschitz Certificates for Layered Network Structures Driven by Averaged Activation Operators SIAM Journal on Mathematics of Data Science, 2(2), 529-557.
> > >
> > > https://doi.org/10.1137/19M1272780
> > >
> > > Abstract
> > >
> > > Obtaining sharp Lipschitz constants for feed-forward neural networks is essential to assess their robustness in the face of perturbations of their inputs. We derive such constants in the context of a general layered network model involving compositions of nonexpansive averaged operators and affine operators. By exploiting this architecture, our analysis finely captures the interactions between the layers, yielding tighter Lipschitz constants than those resulting from the product of individual bounds for groups of layers. The proposed framework is shown to cover in particular many practical instances encountered in feed-forward neural networks. Our Lipschitz constant estimates are further improved in the case of structures employing scalar nonlinear functions, which include standard convolutional networks as special cases.

---

### Meta-Review · Area_Chair_c7h7 · 2022-08-24

**Recommendation:** Accept
**Confidence:** Certain

**Metareview:**

This paper provides an approximation error for Sobolev type functions by using deep CNNs. The approximation error achieves the optimal rate and has adaptivity to the low dimensionality of the support of the input data distribution. They also derive a classification error bound using the approximation error result.
This paper gives a novel and important theoretical result. Due to the CNN structure, it requires a different techniques from that for FNNs and thus the analysis is not trivial. This gives an important contribution to the literature. Thus, I recommend acceptance for this paper.

**Award:**

No

---

### Decision · Program_Chairs · 2022-09-14

Accept